# Electrocatalytic hydrogenation of acetonitrile to ethylamine in acid

Chongyang Tang[1,4], Cong Wei[2,4], Yanyan Fang[2,4], Bo Liu[2], Xianyin Song[1], Zenan Bian[2], Xuanwei Yin[2], Hongbo Wang [1], Zhaohui Liu[2], Gongming Wang [2] ✉, Xiangheng Xiao [1] ✉ & Xiangfeng Duan [3] ✉

Electrochemical hydrogenation of acetonitrile based on well-developed proton exchange membrane electrolyzers holds great promise for practical production of ethylamine. However, the local acidic condition of proton exchange membrane results in severe competitive proton reduction reaction and poor selection toward acetonitrile hydrogenation. Herein, we conduct a systematic study to screen various metallic catalysts and discover Pd/C exhibits a 43.8% ethylamine Faradaic efficiency at the current density of 200 mA cm$^{-2}$ with a specific production rate of 2912.5 mmol g$^{-1}$ h$^{-1}$, which is about an order of magnitude higher than the other screened metal catalysts. Operando characterizations indicate the in-situ formed PdH$_x$ is the active centers for catalytic reaction and the adsorption strength of the *MeCH$_2$NH$_2$ intermediate dictates the catalytic selectivity. More importantly, the theoretical analysis reveals a classic d-band mediated volcano curve to describe the relation between the electronic structures of catalysts and activity, which could provide valuable insights for designing more effective catalysts for electrochemical hydrogenation reactions and beyond.

Ethylamine, a key basic chemical in chemical industry, is extensively used for the synthesis of pharmaceuticals and fine chemicals[1-3]. Hydrogenation of acetonitrile (HAN) through thermocatalytic process with pure H$_2$ as the hydrogen sources is the main industrial method for ethylamine production[4,5]. Such process typically operates under high temperature and pressure, raising severe concerns on safety and economic costs[6-8]. Alternatively, electrocatalytic HAN (E-HAN) using water as the hydrogen sources can be performed at mild condition. The earliest researches on E-HAN can date back to 1980s[9], but were limited by low acetonitrile conversion rates and poor ethylamine selectivity that undermines its competitiveness with conventional thermocatalytic HAN processes[10,11]. Searching new catalysts with favorable selectivity and catalytic activity is intrinsically crucial for E-HAN. Recently, copper-based catalysts were found to be effective in alkaline condition for electrochemical hydrogenation of acetonitrile with 86% ethylamine faradaic efficiency at the current density of 100 mA cm$^{-2}$ [12]. These demonstrated catalytic processes are typically operated in alkaline media with the cell configurations similar to conventional alkaline water electrolyzers. Such configuration basically leads to high polarization loss stemmed from the large reaction overpotential (oxygen evolution reaction on anode and E-HAN on cathode) and high internal ohmic resistance due to the sluggish ion diffusion in alkaline medium[13-15].

Inspired by the advantageous configuration of polymer electrolyte membrane electrolyzer with fast ion diffusion rate (polymeric electrolyte) and low internal ohmic resistance (zero-gap design), E-HAN based on polymer-based membrane electrolysis may offer a more energy-efficient approach. Among the polymeric electrolyte membranes, commercial proton exchange membrane (PEM) has a demonstrated 10-year lifetime, which is practically suitable for electrochemical catalytic process[16-18]. Therefore, it is highly desirable to develop catalytic E-HAN electrolyzer based on PEM. However, the

[1]School of Physics and Technology, Wuhan University, Wuhan, P. R. China. [2]School of Chemistry and Materials Science, University of Science and Technology of China, Hefei, P. R. China. [3]Department of Chemistry and Biochemistry, University of California, Los Angeles, Los Angeles, CA, USA. [4]These authors contributed equally: Chongyang Tang, Cong Wei, Yanyan Fang. ✉e-mail: wanggm@ustc.edu.cn; xxh@whu.edu.cn; xduan@chem.ucla.edu

harsh working condition in PEM system raises great challenges for the catalyst development[19,20]. The local strong acidic environment of Nafion ionomers in PEM system, typically deactivate or destabilize the typical catalysts used in alkaline medium. In addition, the acidic condition may favor hydrogen evolution reaction[21–24], leading to a low selectivity toward ethylamine (Fig. 1). To date, there is limited efforts in developing E-HAN catalysts in acidic condition to the best of our knowledge. In this regard, exploring E-HAN catalysts with acidic activity and high selectivity is not only fundamentally meaningful, but also useful for practical ethylamine production.

Since the competitive adsorption of H and N-containing intermediates[25,26] is closely related the intrinsic electronic structures of catalysts[27–30], screening metallic electrocatalysts to reveal the structure-activity relation is critical. Herein, we conduct a systematic investigation of the catalytic activity of a series of metal catalysts including Pd, Ag, Au, Pt, Cu, W and Mo toward E-HAN in acidic medium. Interestingly, Pd nanoparticles display a maximal FE of -66.1% for ethylamine production with a large partial current density, which is 5.2 times better than the copper catalysts. To minimize the large overpotential loss originated from the 4-electron oxygen evolution reaction (OER), hydrogen oxidation reaction (HOR) anode is used and the assembled PEM catalytic electrolyzer with Pd cathode catalyst and Pt/C dry anode display an onset potential of 0 V, and can achieve a current density of 200 mA cm$^{-2}$ at 0.76 V, with an ethylamine FE of 43.8% and a specific production rate of 2912.5 mmol g$^{-1}$ h$^{-1}$, which is about 10 times higher than the other screened metal catalysts. Operando spectroscopic characterizations and density functional theory (DFT) calculations demonstrate the in situ formed palladium hydride (PdH$_x$) intermediates facilitate the catalytic generation of ethylamine. More importantly, a classic d-band mediated volcano profile is revealed to describe the relation between the electronic structures of metallic catalysts and E-HAN activity. This work provides valuable insight into the fundamental understanding on the catalytic chemistry of E-HAN and beyond.

## Results
### Catalyst screening and performance analysis
The screening of electrocatalysts for acetonitrile reduction in acid condition was conducted by studying seven monometallic catalysts including Pd, Cu, Au, Pt, Ag, W and Mo, which are chemically stable in acid and commonly used in the thermocatalytic hydrogenation of nitrile. These metallic catalysts were supported on carbon black (XC72R) and further confirmed by X-ray diffraction (XRD) patterns and transmission electron microscope (TEM) images (Supplementary Figs. 1–4). The E-HAN activity was first evaluated in a H type-three electrode electrolytic cell. The amount of ethylamine product was quantified by $^1$H nuclear magnetic resonance (NMR) spectroscopy and the gas product, hydrogen (H$_2$), was analyzed by gas chromatography (GC). The Faradaic efficiency (FE) distributions and partial current density were analyzed at different applied potentials ranging from 0 to −0.8 V vs. RHE in 0.5 M H$_2$SO$_4$ electrolyte containing 8 wt% acetonitrile. The maximum ethylamine FE together with the corresponding H$_2$ FE is summarized in Fig. 2a (more details in Supplementary Fig. 5). Among all the metallic catalysts, Pd/C showed the highest ethylamine FE of 66.1%, with an ethylamine partial current density of 288.7 mA cm$^{-2}$, indicating an excellent catalytic selectivity and intrinsic activity (Supplementary Figs. 6 and 7). The Cu/C, Au/C and Pt/C generally exhibit an ethylamine FE < 20% and the partial current density <10 mA cm$^{-2}$. Almost no ethylamine is detected for Ag/C, W/C, and Mo/C, and the major product is H$_2$ in the studied potential regime (relevant characterizations are shown in Supplementary Figs. S8–11). In addition, from the results of the $J$-$t$ curves, all the screened metallic catalysts are relatively stable for short-term studies under the reduction potential in the acidic environment (Supplementary Fig. S8).

Considering Pd possesses the best HAN catalytic performance among the studied metal catalysts, its detailed catalytic properties are further studied. Figure 2b shows the specific FEs of ethylamine, diethylamine, triethylamine and H$_2$ under various applied potentials. In brief, the ethylamine FE increases with increasing overpotential, and reaches 66.1% at −0.57 V vs. RHE with industrial-grade current density (288.7 mA cm$^{-2}$) (Fig. 2c). The sum FE of diethylamine and triethylamine is less than 8%, indicating that electroreduction HAN on Pd/C is highly selective to produce primary amine. The higher ethylamine selectivity in acidic electrocatalytic HAN, is distinct from thermocatalytic hydrogenation route that typically show a much lower selctvity towards ethylamine due to high reaction temperature that promote secondary amine formation.

Based on the screened high-performance Pd/C catalysts in H-type three electrode system, we further applied such catalysts in PEM catalytic electrolyzer and evaluated its potential for practical application at the device level. To minimize the potential loss from the anode side of oxygen evolution reaction (OER), we coupled the cathodic E-HAN reaction with hydrogen oxidation reaction (HOR) in the anode and used Pt/C as the anode catalysts. The schematic illustration of the PEM catalytic electrolyzer is shown in Fig. 2d and Supplementary Figs. 12 and 13. For membrane electrode assembling, the catalyst-coated substrate (CCS) method is utilized (Supplementary Fig. 14). The cathode is fed with 0.5 M H$_2$SO$_4$ containing 8 wt% acetonitrile aqueous solution for ethylamine production, while the Pt/C anode is supplied with humidified H$_2$ to perform the HOR (HOR||HAN). A control experiment is designed with the anodic water oxidation reaction, in which IrO$_2$ is used as OER catalysts (denoted as OER||HAN). The corresponding polarization curves of both HOR||HAN and OER||HAN are shown in Fig. 2e. Apparently, the onset voltage of E-HAN through HOR||HAN is -1.5 V lower than that through OER||HAN. More interestingly, the E-HAN can occur at 0 V by coupling HOR reaction, suggesting the E-HAN reaction can spontaneously proceed in such device. In addition, it only requires cell voltages (E$_{cell}$) of -0.61 V and -0.87 V to achieve a current density of 100 mA cm$^{-2}$ and 200 mA cm$^{-2}$, respectively (Fig. 2f). To quantify the selectivity of the PEM catalytic reactors, the Faradaic efficiencies of the amine products are studied under various current densities. At the current densities at 100 and 200 mA cm$^{-2}$, the ethylamine FE of Pd/C are -63.01% and -43.8%, respectively (Fig. 2f). With current densities continue to increase, a large amount of hydrogen is produced, leading to gradually decreased ethylamine FE. Furthermore, the dependence of acetonitrile concentration is investigated by varying the mass fraction of acetonitrile in the cathode electrolyte through the MEA electrolyzers (Supplementary Fig. 15). As the concentration of acetonitrile increases from 4 to 12 wt%, the ethylamine FE increases. More precisely, at the cell voltage of 0.95 V, the ethylamine partial current density reaches to 132 mA cm$^{-2}$ in 12 wt% acetonitrile, which is

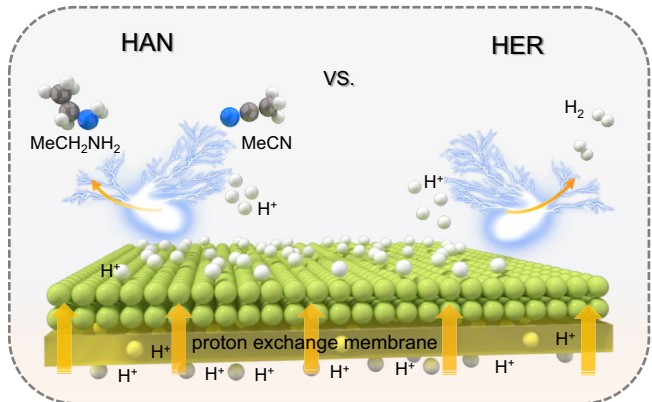

**Fig. 1 | Schematic diagram of ethylamine synthesis.** Schematic illustration of electrocatalytic acetonitrile hydrogeneration in acidic condition, in which hydrogen evolution reaction is the competitive reaction.

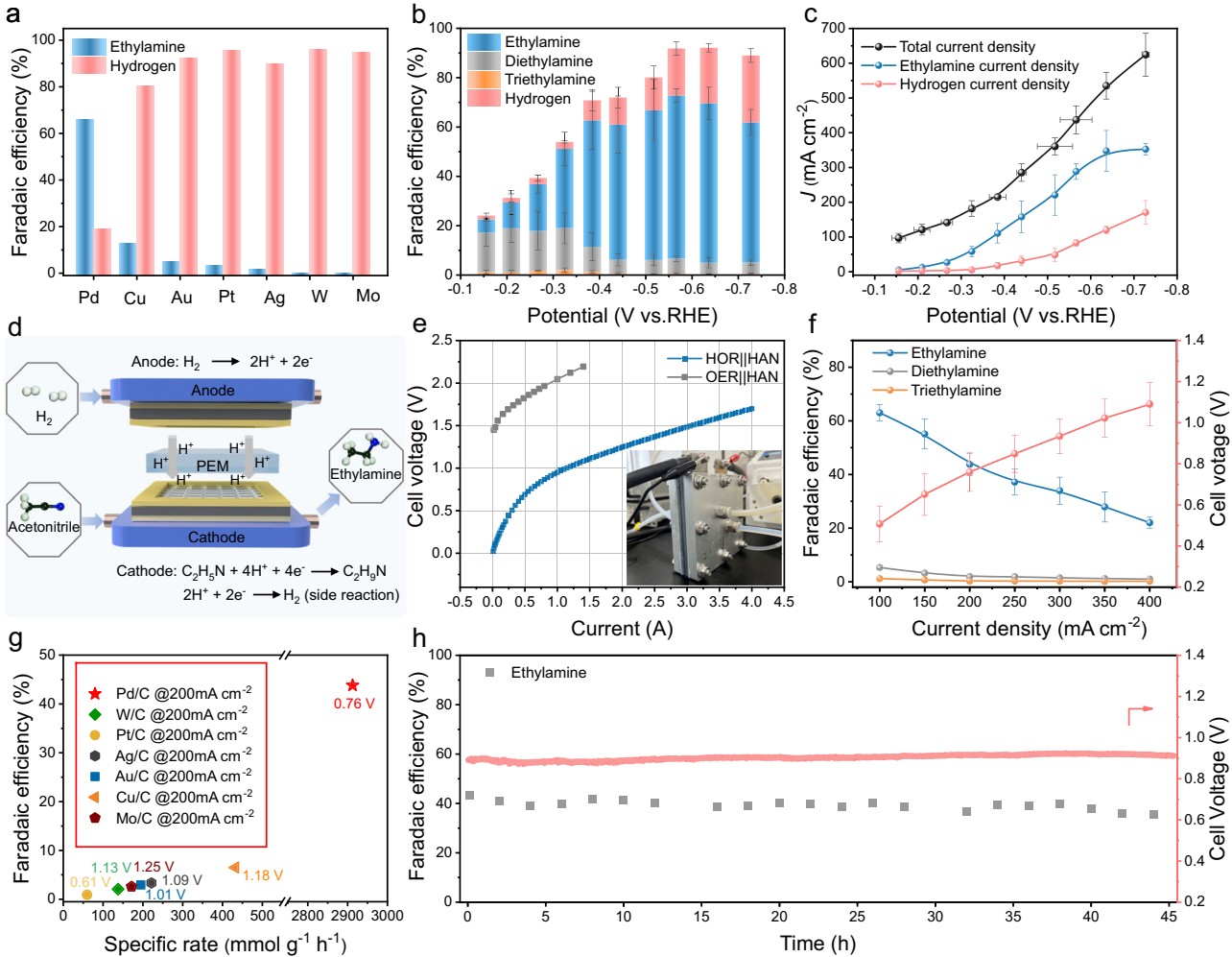

**Fig. 2 | Acetonitrile reduction performance in acidic electrolyte. a** The highest FE of ethylamine on various metal catalysts and corresponding H₂ FE in the applied potential range of −0.15 to −0.75 V versus RHE. **b** Ethylamine FE, diethylamine FE, triethylamine FE, H₂ FE, and **c** corresponding total current density, ethylamine and hydrogen partial current density versus applied potentials on Pd/C. **d** Schematic illustration of the PEM catalytic reactor. **e** Polarization curves test for HAN on Pd NPs cathode and HOR on Pt/C anode or OER on IrO₂ anode. The inset is the PEM catalytic cell picture. **f** Ethylamine, diethylamine FE, and corresponding cell voltage versus applied current on Pd/C. **g** Comparison of the performance of the screened E-HAN catalysts. **h** Stability test over a span of 45 h using Pd NPs as the cathode catalyst at a constant current of 0.8 A. Error bars represent the standard deviation from at least three independent measurements.

1.11 times and 1.18 times higher than those in 8 wt% and 4 wt% acetonitrile, respectively. Furthermore, the other electrocatalysts screened by the H-type three-electrode system were also tested at the current density of 200 mA cm⁻² in PEM electrolyzer (Fig. 2g). The PEM reactor with Pd/C catalyst achieves a specific activity of 2912.5 mmol g⁻¹ h⁻¹ at a relatively low voltage, which is almost an order of magnitude higher than the other studied electrocatalysts, verifying the superior activity of Pd/C for the E-HAN. Preliminary techno-economic analysis is performed to assess economic feasibility. At a typical commercial current density of 200 mA cm⁻², the OER‖HAN yields a negative upcycling net revenues of approximately $−63.85 per ton of acetonitrile (Supplementary Fig. 16a). In contrast, when the anode is substituted with HOR, the upcycling net revenues jumps to around $195.66 per ton (Supplementary Fig. 16b), overweighing the OER‖HAN electrolyzer (Supplementary Fig. 17 and Supplementary Table 1).

The MEA electrolyzers also showed a stable cell voltage (-0.89 V) and a steady ethylamine FE (>35%) at a constant current density of 200 mA cm⁻² for 45 h (Fig. 2h), demonstrating the reasonable stability. The structural stability of the Pd NPs is further examined by transmission electron microscopy (TEM), scanning electron microscopy (SEM), XRD and X-ray photoelectron spectroscopy (XPS) (Supplementary Figs. S18–21), in which no obvious change is observed. The

membrane stability is also evaluated by swelling and ion exchange capacity (IEC) tests. The results show that the change of swelling rate and the loss of -HSO₃ were not obvious during the test process (Supplementary Tables 2 and 3). Taken together, the Pd/C catalysts with high activity and selectivity toward acidic E-HAN system holds great promise for practical production of ethylamine from acetonitrile driven by renewable electricity.

## Operando spectroscopic characterizations

To gain an in-depth understanding of the E-HAN process, we carried out the operando synchrotron-radiation Fourier transform infrared spectroscopy (SR-FTIR) and operando Raman spectroscopy under working conditions, which are powerful to probe surface-adsorbed species (Fig. 3a). Figure 3b shows in situ SR-FTIR spectra of E-HAN on Pd NPs over the potential ranging from 0.1 to −0.75 V vs. RHE. At open circuit potential (OCP) and +0.1 V, weak vibration bands located at -787 and 1015 cm⁻¹, which can be assigned to the vibration mode of δ(C-C≡N) and r(CH₃) in MeCN, respectively, were observed[31]. In addition, the vibration band at 2098 cm⁻¹ originates from the emergence of Pd−H stretching mode[32,33]. Along with increasing potential (more negative than +0.1 V), the vibration intensities of δ(C-C≡N), CH₃ and Pd−H are gradually strengthened, suggesting the gradually increased

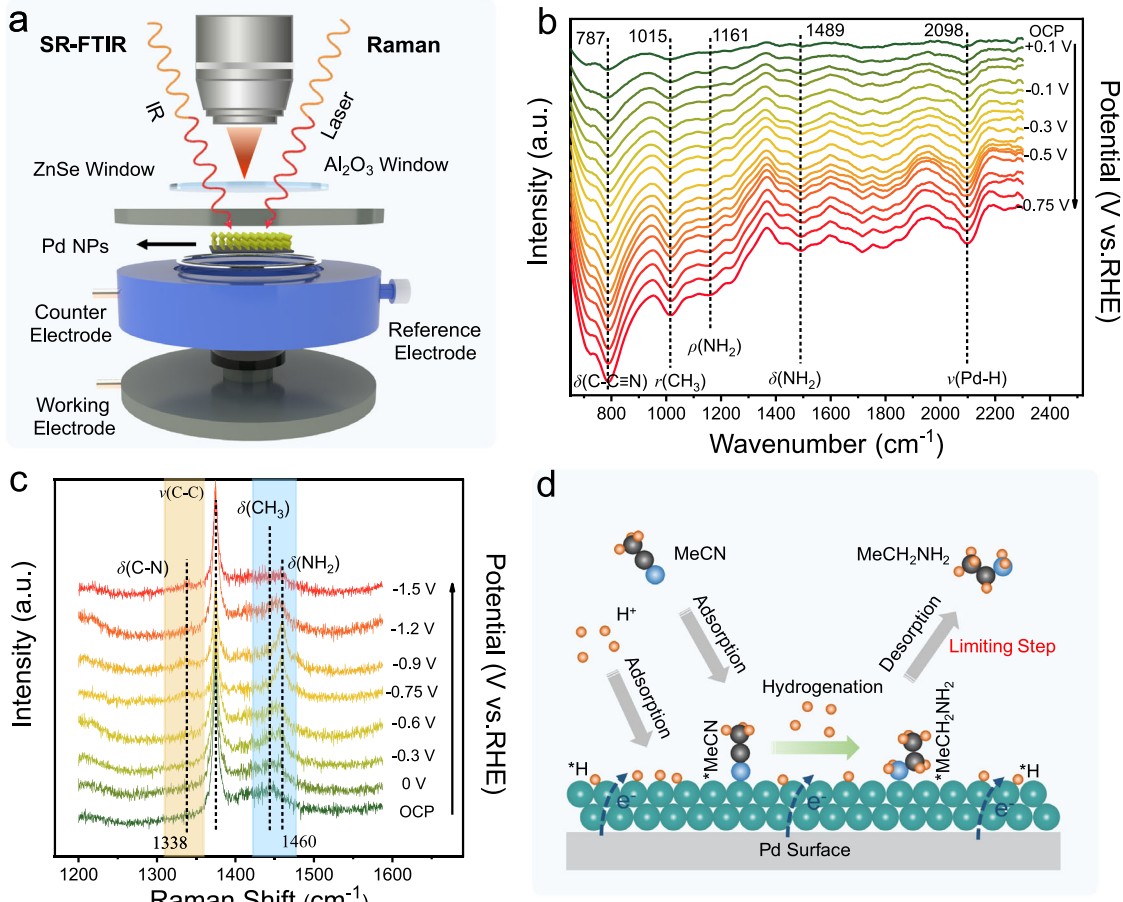

**Fig. 3 | Operando spectroscopic characterizations of the reaction intermediates for Pd catalyst. a** The schematic illustration of operando SR-FTIR and Raman electrolytic cell. **b** The operando SR-FTIR spectra and **c** electrochemical

Raman spectra at potentials vs. RHE. **d** Schematic illustration of the surface adsorption and evolution of the representative intermediates on Pd surface.

surface coverage of MeCN and H on the surface of Pd/C catalyst. At the same time, the rocking mode and bending mode of NH₂ at ~1161 and 1489 cm⁻¹, as the key feature of the formation of ethylamine[34], gradually emerges, implying that the electroreduction of MeCN occurs. In comparison, the operando SR-FTIR spectroscopy of the Cu, Pt, and Au, which has relatively low E-HAN performance, are also studied and summarized in Supplementary Fig. 22. We observe that acetonitrile molecules accumulate to varying extents on different metal surfaces, and this variation is probably associated with the differing strengths of molecular adsorption on these surfaces. The superior performance of Pd catalysts is likely attributable to their capacity for moderate molecular adsorption, which will be substantiated by the results of DFT calculations.

The operando Raman spectra of E-HAN on Pd NPs at different potentials are plotted in Fig. 3c. The Raman bands at 1375 and 1443 cm⁻¹ can be assigned to the C–C stretching modes and C–H bending modes of the MeCN molecules, whereas the band at 1639 cm⁻¹ is attributed to δ(OH) band in H₂O, and those at 1338 and 1460 cm⁻¹ are attributed to the stretching modes of δ(C–N) and the bending mode δ(NH₂) of the amine products[30,35,36], respectively (Fig. 3c and Supplementary Fig. 23). Compared with the spectrum collected at OCP, the peaks of δ(C–N) and δ(NH₂) gradually appear and become strengthened when the potential is below 0 V, suggesting the reduction of MeCN on Pd NPs occurs from 0 V, consistent with the SR-FTIR results. As the cathodic potential becomes more negative, the peak of NH₂ at 1460 cm⁻¹ increases prominently and reaches the maximal value at −0.9 V, while further increasing the cathodic potential will reduce this

peak. The intensity changes of the peak of NH₂ indicate the sorption behavior of MeCH₂NH₂* is strongly related to the electrode potential and the MeCH₂NH₂* is the key intermediate for the formation of MeCH₂NH₂. Combining with the electrochemical results in Fig. 2b, at low cathodic potentials (less negative), the desorption of *MeCH₂NH₂ to ethylamine is hindered and the MeCH₂NH* intermediate is coupled to form diethylamine. When the cathodic potential becomes more negative, *MeCH₂NH₂ desorption becomes easier, leading to the increased selectivity of ethylamine (Fig. 3d). In addition, the vibration band signals of C–N and NH₂ for ethylamine synthesis on Pt, Au, Cu are not detected (Supplementary Fig. 23), which means poor E-HAN activities, in line with the SR-FTIR analysis and electrochemical test results.

Operando X-ray absorption fine structure (XAFS) experiments are performed to investigate the electronic and geometric properties of the Pd/C catalysts during E-HAN process (Fig. 4a). Before the XAFS experiments, the sample was pre-activated in 0.5 M H₂SO₄ at 0 V vs. RHE for 30 min to avoid mild oxidation of Pd catalysts in air. To investigate the local environment variations around Pd NPs during HAN process, the Fourier-transformed extended XAFS (EXAFS) spectra and fitting spectra in the R-space were performed by using the Demeter software package. The k²-weighted spectra were subjected to Fourier transformation in the k-range 2.4−11 Å and the first Pd–Pd shell of the Fourier-transformed EXAFS spectra was fitted by using an amplitude reduction factor of 0.816 that was obtained from the Pd foil (Supplementary Table 4 and 5). Figures 4b and 4c show the operando EXAFS spectra and the corresponding fitting results obtained in 0.5 M

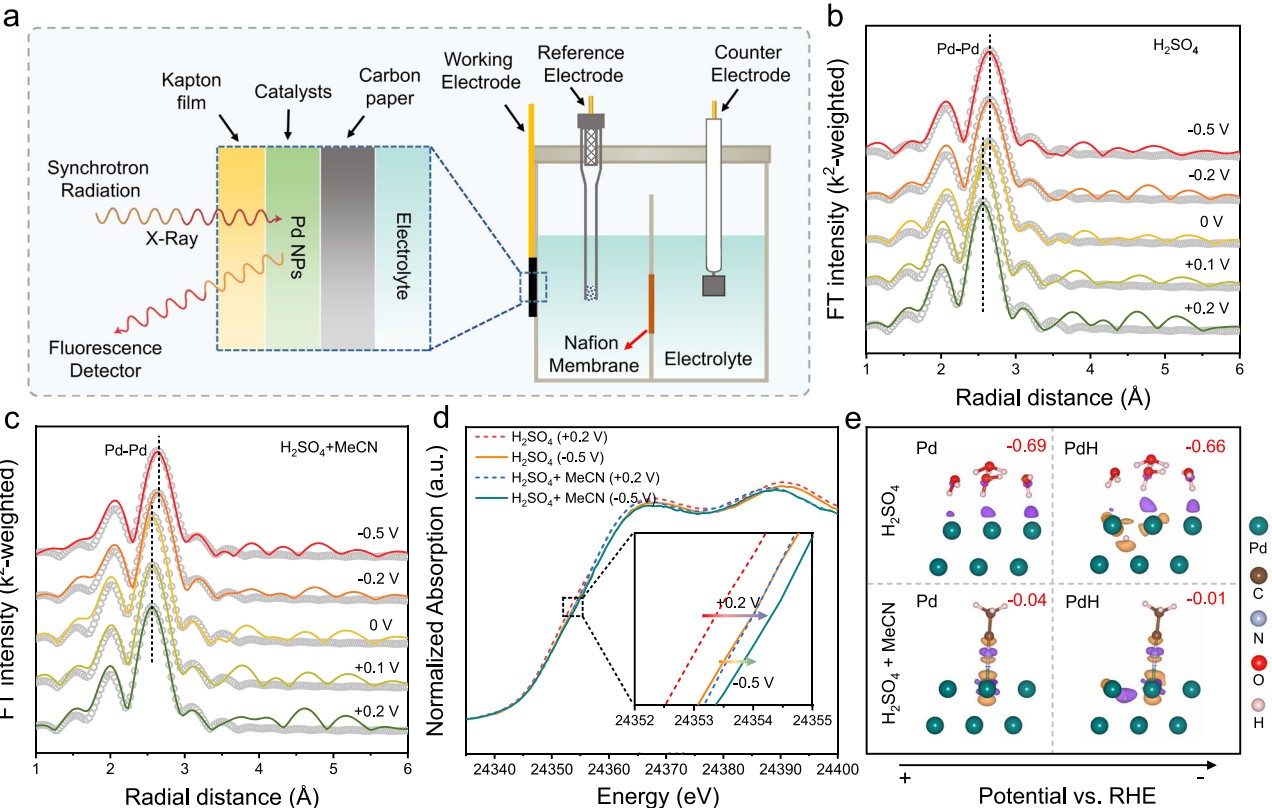

**Fig. 4 | Operando XANES spectroscopy measurement results. a** The schematic illustration of operando XANES electrolytic cell. Potential dependence of operando EXAFS spectra of Pd K-edge in 0.5 M $H_2SO_4$ solution (**b**) and 8 wt% acetonitrile in 0.5 M $H_2SO_4$ solution (**c**). **d** Potential dependence of operando Pd K-edge XANES spectra of the catalyst in 0.5 M $H_2SO_4$ solution and after the addition of 8 wt% MeCN

at +0.2 and −0.5 V vs. RHE. **e** Charge density difference plot for $H_9O_4$ and MeCN adsorbed on Pd and PdH. Charge accumulation and depletion regions are shown in purple and yellow, respectively, with an iso-surface value of 0.005 e $Å^{-3}$. Red numbers in top right corner show Bader charge transfer from the molecular to the Pd sites.

$H_2SO_4$ without and with 8 wt% MeCN additive. The EXAFS results in the 0.5 M $H_2SO_4$ electrolyte reveal that the bond length of Pd−Pd increases gradually from ~2.73 (at +0.2 V RHE) to ~2.80 Å (at 0 V RHE) and remains almost constant with further increasing the cathodic potential. The volume expansion that might originate from the hydride formation ($PdH_x$), which is a typical phenomenon for Pd[37,38]. It indicates, at the potential window (<0 V), the real reactive species is the $PdH_x$ rather than pure metallic Pd. Meanwhile, for the EXAFS spectra collected with MeCN additive, the potential at which Pd is fully hydrogenated to the stable state (from 2.73 to 2.78 Å) is reduced to −0.1 V, indicates the formation of $PdH_x$ is slightly hindered. Considering the transformation of Pd to $PdH_x$ is related to the diffusion of surface adsorbed proton to the Pd lattice, the hindered $PdH_x$ formation with the presence of MeCN might originate from the suppressed H adsorption by the competitive adsorption of MeCN. However, at negative potentials, the lattice expansion also suggests the real catalytic species is $PdH_x$ during E-HAN.

Furthermore, the electronic structure change is investigated by the X-ray absorption near edge structure (XANES). The profile of XANES spectra of the Pd K-edge at cathodic potential of −0.5 V vs. RHE in 0.5 M $H_2SO_4$ electrolyte have a shift to a higher energy in contrast to that of Pd K-edge at +0.2 V vs. RHE, which is related to $PdH_x$ formation with potential change (Fig. 4d and Supplementary Fig. 24), as demonstrated by the differential charge density and Bader charge analysis in Fig. 4e and Supplementary Figs. 25 and 26. The differential charge density of the hydrated proton adsorption on $PdH_x$ structure shows that Pd sites around H have a charge depletion, as proved by Bader charge gain of Pd from 0.69 e⁻ to 0.66 e⁻. After adding 8 wt%

MeCN into the electrolyte, the both absorption edges at +0.2 V and −0.5 V shift to higher energy. Considering no Faradaic process happens at +0.2 V, the blue shift of the absorption edge after adding MeCN might be attributed to the MeCN adsorption. As indicated by the differential charge density and Bader charge analysis, after the adsorption of MeCN on Pd surface, owing to the electron transfers from Pd to anti-bonding π orbital of the adsorbed MeCN (*MeCN), the Bader charge of Pd is decreased from 0.69 e to 0.04 e, resulting in higher valence state. In a word, both the formation of $PdH_x$ and the strong adsorption of MeCN would increase the valence state of Pd, which is consistent with the operando XAS results.

## Discussion

To correlate the acetonitrile hydrogenation activity trends of the studied metal (Ag, Au, Cu, Pt, Pd, W, Mo) catalysts with their intrinsic electronic structures, theoretical analysis was conducted. It is worth noting that $PdH_x$ is also considered because $PdH_x$ phase is formed during the cathodic reaction based on the in situ experimental characterizations. Considering that catalytic activity is closely related to interfacial electronic coupling, the binding strength of the reaction intermediates is a good descriptor for identifying catalytic activity of surface reaction. Therefore, the binding information of the intermediates involved in the E-HAN on several transition metal surface (Ag (111), Au (111), Cu (111), Pt (111), Pd (111), $PdH_x$ (111), W (110) and Mo (110) surfaces) were investigated (Fig. 5a). It is found that the binding strength generally follows the order W (110) > Mo (110) > Pd (111) > $PdH_x$ (111) > Pt (111) > Cu (111) > Au (111) > Ag (111) for the intermediates that bind via N, hydrogenated N (i.e., $NH_x$) and or C/$CH_x$

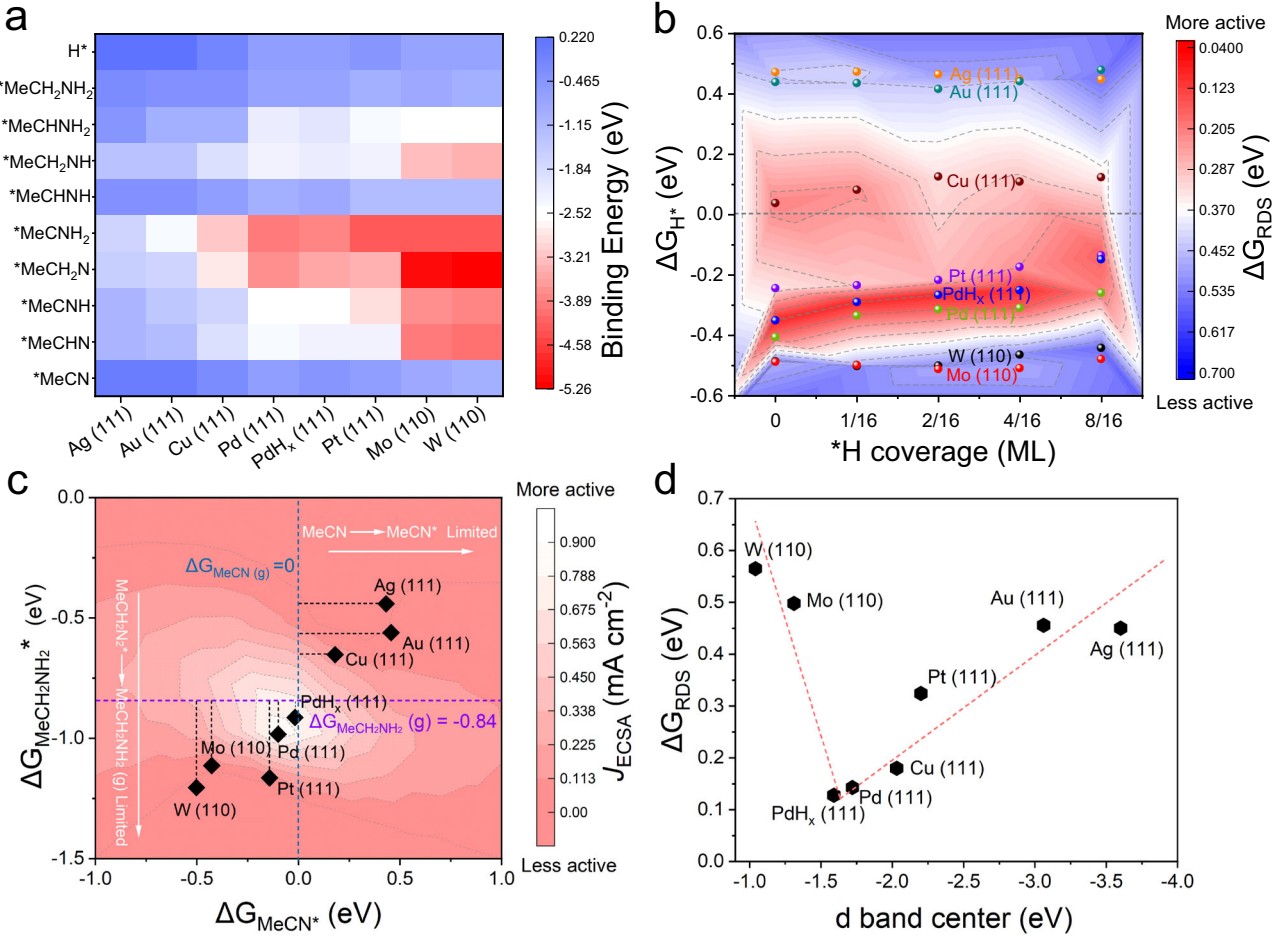

**Fig. 5 | Theoretical analysis for acetonitrile hydrogenation. a** Binding energy of adsorbates without H coverage on Ag (111), Au (111), Cu (111), Pt (111), Pd (111), PdH$_x$ (111), W (110) and Mo (110) surfaces. **b** The HAN reaction free energy change for rate determining steps (ΔG$_{RDS}$) and free energies of *H absorption (ΔG$_{H*}$) on the modeled surface of Ag (111), Au (111), Cu (111), Pt (111), Pd (111), PdH$_x$ (111), W (110) and Mo (110) under various *H coverages. **c** Activity profile versus both ΔG$_{MeCN}$ and ΔG$_{MeCH2NH2}$. Rate map for acetonitrile hydrogenation obtained from *H coverage of 0 ML. The current densities normalized by the ECSA are obtained in experiments at −0.375 V vs. RHE. **d** Relationship between ΔG$_{RDS}$ and metal d-band center.

(Supplementary Figs. 27–29 and Supplementary Table 6). Obviously, the binding strength of Pd (111) and PdH$_x$ (111) is neither too strong nor too weak, which might be favorable for catalysis based on Sabatier principle[39,40].

The free energy changes were further calculated to evaluate thermodynamic barrier for the HAN along four potential pathways shown in Supplementary Fig. 29 using the computational hydrogen electrode mode. A comparison of free energy diagrams calculated at an applied potential U = 0 V along the most favorable pathways demonstrates that acetonitrile hydrogenation is thermodynamically favorable on Pd (111) (0.143 eV) and PdH$_x$ (111) (0.04 eV) compared to Ag (111), Au (111), Cu (111), Pt (111), W (110) and Mo (110) (Fig. 5b and Supplementary Fig. 30), which could be concluded based on the calculated the reaction free energy change for rate determining steps (ΔG$_{RDS}$). It is worth noting that HER is the main competing reaction in the electrochemical HAN reaction under acid condition. Interestingly, the H adsorption free energy of HER on Pd (111) and PdH$_x$ (111) is as high as −0.41 and −0.35 eV. The results suggest the acetonitrile hydrogenation reaction is energetically more favorable than the HER on Pd (111) and PdH$_x$ (111), which eventually benefits the HAN rather than HER to H$_2$ for Pd catalysts, accounting for the experiment results. Based on the results of operando SR-FTIR, and considering the H* would accumulate on metal surface with the increase of cathodic potential, the influence of different surface *H coverages (i.e., 1/16 ML, 2/16 ML, 4/

16 ML, and 8/16 ML) on the HER and HAN activity was explored in the calculations as shown in Fig. 5b and Supplementary Figs. 31–41. It is found that acetonitrile hydrogenation activity of PdH$_x$ was consistently the highest among the several metal surfaces considered, ranging from low *H coverage (0/16 ML) to medium *H coverage (4/16 ML), as shown in Fig. 5b. As the H coverage continued to increase to 8–16 ML, the value of ΔG$_{RDS}$ instead increased from 0.04 eV (4–16 ML) to 0.12 eV, which was less than G$_{H*}$ (0.15 eV), implying that HER side effects could be dominated. That is to say, with the decrease of the cathodic potential, the evolution of Pd into PdH$_x$ increase the selectivity of acetonitrile hydrogenation, but as the potential continues to decrease, the hydrogen coverage will increase, which leads to a decrease in the selectivity of acetonitrile hydrogenation.

To search a key descriptor for the HAN activity, we analyzed and summarized the free energy diagrams of all metal surfaces at H coverage of 0 ML, and then consolidate the mechanistic insights into a general kinetic activity volcano for acetonitrile hydrogenation determined by two activity descriptors, the free energy of adsorption of MeCN, ΔG$_{MeCN}$, and that of MeCH$_2$NH$_2$, ΔG$_{MeCH2NH2}$ (Fig. 5c). The theoretical maximum in ECSA normalized activity occurs at intermediate ΔG$_{MeCN}$ = 0 and at ΔG$_{MeCH2NH2}$ = −0.84 eV, where both the MeCN formation and MeCH$_2$NH$_2$ desorption steps are facile. It allows us to determine which step is rate limiting in acetonitrile hydrogenation to ethylamine, using the computed MeCN*, MeCH$_2$NH$_2$* free

energies. It is noteworthy that PdH$_x$ and Pd are located closest to the intersection of the two lines among all surfaces, correlates well with the trends in the HAN activity (i.e., Current density normalized by ECSA at $-0.375$ V vs. RHE) observed in experiments. This means that on a given surface, the activity can be roughly inferred by counting only $\Delta G_{MeCN}$ and $\Delta G_{MeCH2NH2}$. In order to further understand the activity of HAN at the level of electronic structure, the d-band centers of several metals have been calculated. Plotting the $\Delta G_{RDS}$ as a function of the d-band center yields an inverted volcano curve (Fig. 5d and Supplementary Fig. 42), on which the most active surface was found to be PdH$_x$ (111), with a moderate d-band center. Based on activity volcano and electronic structure arguments, we can conclude the ideal catalyst should have suitable d-band center with empty orbital to stabilize MeCN* but can't overhybridize with MeCH$_2$NH$_2$* to form poison species, such that MeCN adsorption and MeCH$_2$NH$_2$ desorption does not require a significant overpotential. Overall, the DFT results are in excellent agreement with the experimental observations and suggest that the formation of PdH$_x$ by cathodic potential promotes acetonitrile electroreduction due to the moderate binding affinity toward the reaction intermediates.

In summary, we have demonstrated electrocatalytic HAN using PEM system with Pd/C as the hydrogenation catalyst to produce ethylamine under ambient conditions. The assembled PEM catalytic electrolyzer exhibits a 43.8% ethylamine FE with a specific production rate of 2912.5 mmol g$^{-1}$ h$^{-1}$ at the current density of 200 mA cm$^{-2}$, which is about an order of magnitude higher than the other screened catalysts. Operando spectroscopic characterizations combined with DFT calculations demonstrate the in situ formed PdH$_x$ promotes the desorption of ethylamine and suppresses the competing HER, which together contribute to the superior E-HAN catalytic activity. Furthermore, a d-band mediated volcano curve is revealed to describe the relation between the electronic structures of catalysts and activity, which could provide useful guidance for designing electrochemical acidic HAN catalysts and beyond.

## Methods

### Materials synthesis

To prepare the Pd NPs, 10 mg of Pd (acac)$_2$ (Shanghai Macklin Biochemical, ≥99.5%), 50 µL of TOP (Sigma-Aldrich, 97% purity), 4 ml of oleylamine (Sigma-Aldrich, 70% purity) and 1 ml of oleic acid (Sigma-Aldrich, ≥99% purity) were mixed in a 20-mL glass vial, and subsequently ultrasonicated for 0.5 h to yield a homogeneous light-yellow solution. The vial was then transferred into an oil bath at 160 °C for 5 h. After cooling to room temperature, the colloidal product was collected by centrifugation (10,000 r.p.m.), and then washed several times with a mixture of cyclohexane and ethanol (2:8) to remove the excess solvent. To synthesis of Pd/C catalysts, as prepared Pd NPs was mixed with 8 mg commercial carbon (Vulcan XC-72) in 1 mL of cyclohexane and 8 mL of ethanol. After sonicating for 1 h, the products were collected by centrifugation (12,000 r.p.m.) and dried naturally.

To synthesize Au NPs, 25 mL of tannic acid ($1.4 \times 10^{-4}$ M) (Shanghai Macklin Biochemical, ≥98%) and 500 µL of 88.5 mM HAuCl$_4$·4H$_2$O (Sigma-Aldrich, ACS reagent, ≥99.9% purity) were mixed in a 50-mL beaker. Subsequently, 1 mL of NaBH$_4$ (Shanghai Sinopharm Chemical Reagent Co., 96% purity) solution (2.5 mg/mL) was added to the above solution under magnetic stirring to obtain a wine-red solution. The Ag was prepared according to the same protocol as that of Au, except that 500 µL of 29 mM AgNO$_3$ (Sigma-Aldrich, ACS reagent, ≥99.8% purity) and 340 µL of NaBH$_4$ (2.5 mg/mL) were used. As for Pt NPs, 25 mL of tannic acid ($1.4 \times 10^{-4}$ M) and 500 µL of 49 mM H$_2$PtCl$_6$ (Sigma-Aldrich, ACS reagent, ≥99.9% purity) were mixed in a 50-mL beaker. Subsequently, 4 mL of NaBH$_4$ solution (2.5 mg/mL) was added to the above solution under magnetic stirring. When the solution became black, it was further kept for 3 h. Finally, the prepared Au NPs, Ag NPs, and Pt NPs was mixed with 20.3, 3.7, and 11.1 mg of commercial carbon

(Vulcan XC-72) in DI water (25 mL) and ethanol (10 mL) to obtain Au/C, Ag/C and Pt/C preparing, respectively. To synthesis of Cu/C, W/C, and Mo/C catalysts, 3.4 mg of as purchased Cu (Alfa Aesar, 99.99%), W (Alfa Aesar, 99.99%), and Mo (Alfa Aesar, 99.99%) were mixed with 8 mg of commercial carbon (Vulcan XC-72) in ethanol (1 mL) and acetone (8 mL). After sonicating for 1 h, the products were collected by centrifugation (12,000 r.p.m.) and dried naturally.

### Material characterization

The TEM images were conducted on a HT-7700 (HT) at an acceleration voltage of 200 kV. Scanning electron micrographs (SEMs) were collected by JEOL S-4800. X-ray diffraction (XRD) was obtained on Bruker AXS, D8 Advance X-ray powder diffractometer with Cu-Kα radiation (λ = 0.15418 nm). H nuclear magnetic resonance (NMR, Agilent 400 MHz) was carried out to qualitatively and quantitatively detect amine products. Operando Raman was measured using a commercial Raman microscope (HR800, Horiba) and a laser emitting at 532 nm was served as the excitation source. Operando synchrotron radiation FTIR measurements were made at the infrared beamline BL01B of National Synchrotron Radiation Laboratory (NSRL, China) through a homemade top-plate cell reflection IR setup with a ZnSe crystal as the infrared transmission window (cutoff energy of -625 cm$^{-1}$). The Operando synchrotron radiation XAS spectra of Pd K-edge was collected in a fluorescent mode, using ionization chambers with optimized detecting gases to measure the radiation intensity, at BL14W1 station in Shanghai Synchrotron Radiation Facility (SSRF).

### Electrochemical measurements

For the preparation of working electrodes, 2 mg of catalyst powder was dispersed in 1 ml of 1:3 v/v isopropanol/DIW mixture with 10 µl of Nafion solution (5 wt%). After sociation for 20 min, the catalyst ink was dropped on Sinero carbon paper (YLS-30T) and keep the areal mass loading and metal mass loading were 1 mg cm$^{-2}$ and 0.3 mg cm$^{-2}$, respectively (geometric surface area, 0.5 cm$^2$). The electrochemical characterizations were carried out in a conventional three-electrode cell at ambient temperature connected to a CHI 760E electrochemical workstation. Leak-free Ag/AgCl and platinum plate were used as reference and counter electrodes, respectively. In a typical E-HAN test, the catholyte was 0.5 M H$_2$SO$_4$ containing 8 wt% acetonitrile aqueous solution, and the anolyte was 0.5 M H$_2$SO$_4$ aqueous solution. 70% iR compensation was applied for potential to compensate for the effect of solution resistance.

For each potential test of Pd NPs, a newly prepared working electrode was used and subjected to electrolysis for 900 s to collect liquid product and effluent gas from the H cell went through the sampling loop of a gas chromatograph (GC) and was analyzed in 5 min intervals to determine the concentration of H$_2$ products. The GC system (GC9790Plus) was equipped with hayesep D column with Ar (Praxair, 5.0 Ultra high purity) flowing as a carrier gas and 5 A columns that connected to a thermal conductivity detector and a flame ionization detector. Liquid products were identified afterwards through a Bruker AVIII 600 MHz NMR spectrometer. In short, 350 µl of the liquid product was added with 150 µl of 3 M KOH and 200 µl of internal standard solution that consisted of 25 ppm (v/v) dimethyl sulfoxide (≥99.9% (Alfa Aesar)) in D$_2$O. Solvent presaturation technique water suppression was applied to analyze the one-dimensional $^1$H spectrum.

### MEA preparation and evaluation

The cathode catalyst ink was prepared by mixing 30 mg of Pd/C with 0.8 mL of deionized water, 2.4 mL of isopropyl alcohol, 108 mg of 5 wt% nafion solution, and 10 mg of 60% PTFE, then ultra-sonicating for 30 min. The anodic catalyst ink was prepared in a similar way as the cathode, except that the amount of Nafion solution is changed to 216 mg. The anode catalyst layer was prepared by spraying the anode Pt/C catalyst ink on the membrane.

The cathode catalyst layer was prepared by spraying the metal catalyst ink on the carbon paper for CCS method and on the membrane for catalyst-coated membrane (CCM) method. The MEA was prepared by sandwiching the electrodes and the PEM between flow channels. The side of the carbon paper sprayed with Pd/C catalyst should not be adjacent to the PEM during the CCM method assembly in the cathode. The catholyte was 0.5 M $H_2SO_4$ containing 8 wt% acetonitrile aqueous solution, and the anode was supplied with humidified $H_2$. The reaction was carried out at room temperature and the flow rate of cathode electrolyte was maintained at 2 min/mL. The area of MEA and areal mass loading was 4 cm$^2$ and 1 mg cm$^{-2}$.

## Theoretical calculations

The calculations were carried out using periodic density functional theory (DFT) with the Vienna Ab-initio Simulation Package (VASP)[41,42]. The projector-augmented wave (PAW) method and plane-wave basis functions were employed to expand the atomic core and valence electrons with kinetic energy cutoff set to 450 eV. The generalized gradient approximation (GGA) with the Predew-Burke-Ernzerhof (PBE) formalism exchange-correlation functional was used to account for core-valence interaction[43]. The reciprocal space was sampled using a Γ-centered Monkhorst-Pack scheme with a 3 × 3 × 1 for surfaces. All the geometry optimization would be converged when the energy difference was smaller than $1.0 \times 10^{-5}$ eV and the Hellman–Feynman force on each ion was smaller than 0.02 eV/Å. The Pt (111), Pd (111), Au (111), Ag (111), W (110), and Mo (110) surfaces were modeled by a periodic four-layer slab repeated in super cell of lateral size 4 × 4 with a vacuum layer of 15 Å between the periodic slabs along the z direction to minimize the artificial interactions. The Pd-terminated $PdH_x$ (111) was modeled using the NaCl ($L_{12}$) crystal structure. The $PdH_x$ (111) surfaces were modeled with 4 × 4 surface slabs consisting of two bilayers and two single layers (a bilayer contains a unit of one Pd layer and one H layer). During geometry relaxation, metal model atoms in the bottom two metal layers and $PdH_x$ (111) model atoms in the bottom three metal layers were fixed while all other atoms were allowed to relax.

The binding energies (BE) were calculated by the following equation:

$$BE_{ads} = E_{adsorbate} - E_A - E_B \quad (1)$$

where $E_{adsorbate}$, $E_A$, and $E_B$ represent the total energy of the surface with the adsorbate, the surface without the adsorbate and the molecular, respectively.

The adsorption Gibbs free energy is determined by the expression:

$$G_{ads} = E_{ads} + ZPE - TS \quad (2)$$

where $E_{ads}$, ZPE, T, and S are total energy obtained from DFT calculations, zero-point energy, entropy, and temperature, respectively, at 298.15 K. For molecules, those were taken from the NIST database. For absorbates, ZPE and S were determined by vibrational frequencies calculations, where all 3 N degrees of freedom were treated as harmonic vibrational motions without considering contributions from the slab.

## Data availability

The data generated or analyzed during this study are included in this published article and its Supplementary information files. Source data are provided with this paper.

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

## Acknowledgements

This work was financially supported by the National Natural Science Foundation of China (Nos. 12025503, 22175163, U23B2072), the National Key Research and Development Program of China (2021YFA1500400).

## Author contributions

G.M.W. and X.H.X. conceived the project; C.Y.T., C.W. and Y.Y.F. conducted the project and contributed equally to this work. C.Y. T. conducted the most of experiments and performed the DFT calculations. C.Y.T., C.W. and Y.Y.F. performed the in situ electrochemical experiments; G.M.W., X.H.X. and C.Y.T. co-wrote the paper. X.F.D. provided some important and constructive suggestions to this work. B.L., X.Y.S., Z.N.B., X.W.Y., H.B.W. and Z.H.L. participated in the various aspects of the experiments and discussions.

## Competing interests

The authors declare no competing interests.
