## [Peer Review File · Nature Communications]

Electrocatalytic hydrogenation of acetonitrile to ethylamine in acidREVIEWER COMMENTS

Reviewer #1 (Remarks to the Author):

This paper presents an investigation into the electrocatalytic hydrogenation of acetonitrile for ethylamine production in acidic condition using PEM electrolyzers. The authors explore various metallic catalysts and find that Pd/C catalysts exhibit high selectivity (94.4%) and activity (specific production rate of 2762.9 mmol g⁻¹ h⁻¹) at a current density of 200 mA cm⁻², outperforming thermocatalytic processes by two orders of magnitude. While the topic is of interest, there are several concerns and limitations that prevent the paper from being qualified for publication in Nature communications.

1. One major concern is that the authors propose using proton exchange membrane-based MEA electrolyzers for acetonitrile electroreduction, focusing on acetonitrile electroreduction in acidic conditions. However, the performance of the Pd/C catalysts reported in this paper does not show good performance in PEM-based MEA electrolyzers compared to previous study using Cu catalysts (J. Mater. Chem. A, 2023, 11, 2210-2217, Chem Catalysis 1.2 (2021): 393-406). The authors demonstrate ~70% ethylamine faradaic efficiency (FE) in acidic conditions using Pd/C in an H-type batch cell. However, in a PEM-based MEA electrolyzer, the ethylamine FE drops to 40%, suggesting that pH is not the sole factor determining performance in PEM-based MEA electrolyzers. The microenvironment, including the impact of the PFSA, mass transportation, product crossover, etc., are also important but have not been comprehensively studied. Additionally, the type of membrane used in this study is not reported, and its stability in the presence of acetonitrile and ethylamine should be investigated. Traditional PEM membranes, such as Nafion, composed of perfluorosulfonic acid, exhibit good stability in water electrolyzers but might not be stable in the presence of organic solvents. Given the strong alkalinity of ethylamine, it is essential to investigate the membrane's stability in extended stability studies. Furthermore, considering the low faradaic efficiency of acetonitrile in acidic conditions, it might be more sensible to use an anion exchange membrane-based MEA electrolyzer capable of running acetonitrile electroreduction in neutral or alkaline conditions. The authors should provide a clearer explanation of how their work extends the existing knowledge and contributes to the field.

2. Although the authors report a 94% selectivity towards ethylamine, the faradaic efficiency is very low (<40%) at a current density of 200 mA cm⁻², compared to previous reports (>90%), indicating inefficient utilization of electrons. Moreover, performing hydrogen oxidation at the anode to provide protons decreases the cell voltage but compromises the advantage of using water as a proton source in electrochemical hydrogenation. Considering the low faradaic efficiency of acetonitrile electroreduction reported in this paper, where 60% of hydrogen consumed at the anode does not contribute to ethylamine production, there is a 1.5 times higher hydrogen consumption compared to traditional thermocatalytic acetonitrile hydrogenation using hydrogen as a proton source. Additionally, it is confusing to report selectivity and faradaic efficiency separately. I would suggest that the authors clarify that selectivity refers to selectivity towards primary amine and report the faradaic efficiency simultaneously.

3. The authors suggest in the SR-FTIR study that the vibration band intensities of CCN and CH₃ for

ethylamine synthesis on Cu, Pt, and Au surfaces are relatively lower than those on Pd, indicating that MeCN molecules are less likely to be adsorbed and activated on Cu, Pt, and Au surfaces compared to Pd. However, SR-FTIR cannot quantitatively compare adsorption energy across different samples, so drawing conclusions based solely on band intensities is not appropriate. To accurately assess the adsorption energy differences on various catalysts, direct experimental characterization methods should be employed.

4. The paper compares the activity of acetonitrile electroreduction with conventional thermocatalytic acetonitrile hydrogenation and claims a production rate of 2762.9 mmol g⁻¹ h⁻¹ using Pd/C catalysts, outperforming thermocatalytic processes by two orders of magnitude. However, comparing mass activity between thermocatalysis and electrocatalysis is unfair, considering the small amount of electrocatalysts used. It would be more comprehensive to evaluate the performance of Pd/C catalysts by comparing them with other electrocatalysts only.

5. I recommend that the authors report the current density instead of the total current in Figure 3 and specify the active area of the electrode.

Overall, while the topic of electrochemical hydrogenation of acetonitrile for ethylamine production is of interest, the paper does not meet the standards required for publication in Nature communications. The authors should address the mentioned concerns, provide additional investigation on the microenvironmental impact in the PEM-based MEA electrolyzer, clarify the stability of the membrane in organic solvents, refine the comparisons made, and improve the reporting of experimental data to strengthen the manuscript.

Reviewer #2 (Remarks to the Author):

Wang, Xiao, Duan and colleagues reported the electrochemical hydrogenation of acetonitrile (EHAN) to ethylamine using a PEM electrolyzer. PEM electrolyzer features zero-gap configuration of electrodes and no added-electrolyte in solution, thus holding a promise in the industrial electrochemical production of chemicals. While a recent report by Jiao and co-workers revealed that EHAN in alkaline media using Cu catalyst is a promising technology, the present work featured some advantages such as small ohmic loss and larger current density. This work seems to propose a promising technology to electrify the ethylamine industry. However, some serious scientific concerns listed below should be addressed prior to further consideration.

– The catalyst screening is problematic. Firstly, the preparation method for Cu/C, W/C, and Mo/C largely differs from that of Pt/C, Au/C and Ag/C. More precisely, Pt/C, Au/C and Ag/C were prepared by chemical reduction of corresponding salts in the presence of carbon support, while Cu/C, W/C, and Mo/C were prepared by physically mixing carbon and metal powder. Thus, the nature of these catalysts is considerably different. Hence, the reviewer does not believe that it is scientifically not appropriate to compare the reactivity of this catalyst. Furthermore, it is misleading to discuss experimental catalytic activity and computational results in parallel, since the nature of catalysts is, again, very different.

- For the catalyst screening, the authors should disclose V-t curve for the experiment. The reviewer is wondering if it is possible to use non-precious metals such as Cu in acidic media. If metal catalysts are dissolving, the potential plausibly dramatically changes during the electrolysis. If a non-precious metal catalyst does not survive in this media, then further discussion on the catalyst activity (such as computations) should be avoided.
- The author did not find NMR data for catalyst screening. Also, TEM is not shown for catalysts other than Pd/C. These data should be disclosed.
- There is no information on the conversion and the yield of EHAN process presented herein. These aspects should also be discussed for a fair comparison with preceding systems.
- The present system uses hydrogen oxidation reaction as an anodic reaction. The reviewer believes the advantage of PEM cell is the use of water as a source of protons and electrons. If hydrogen gas is used, then the process requires H₂ tank, making the process much more complicated. Also, H₂ gas is currently produced by steam reforming of methane, which causes an increase in CO₂ emission. Also, the current efficiency is limited to 35% for 20 h electrolysis as shown in Figure 3e, which means 65% of H₂ gas is basically wasted. The authors should make some comments on this point, and discuss if this process can still compete with the thermal process.
- The authors should do some calculations on how much it cost to produce ethylamine using this process.
- Operando IR in PEM cell using Pd/C catalyst is recently reported by Kondo and Atobe, where on top Pd-H species was successfully observed for the first time (DOI: 10.1021/acs.jpcc.2c05127). It would be informative to discuss such on top Pd-H species in this system.
- The use of PEM electrolyzer for the production of fine chemicals is gaining a high level of attention. Especially, the group of Atobe and Shida demonstrated a variety of reactions using PEM electrolyzer. Their manuscripts should be properly cited.
Representative Articles:
DOI: 10.1021/acsenergylett.2c02573
DOI: 10.1021/acscatal.2c01594
DOI: 10.1021/acssuschemeng.9b01882

Reviewer #3 (Remarks to the Author):

Electrochemical hydrogenation of acetonitrile based on acid proton exchange membrane electrolyzers still faces huge challenge because the local acidic condition of PEM results in severe competitive proton

reduction reaction and poor selectivity toward E-HAN. Here, the authors conduct a systematic study to screen various metallic catalysts and discover Pd/C catalysts exhibit good ethylamine selectivity at the high current density, the results are very interesting, However, the research system lacks innovation for Nat. Commun, and there are still some problems to be solved :

(1) The different catalysts are not easy to compare due to some structural differences of catalysts, like the size, morphology, active surface area etc, the comparison of the activity of different catalysts is unscientific.

(2) The ethylamine selectivity of 94.4% is easy to be confused because it doesn't contain hydrogen evolution.

(3) The Pd nanoparticles display a maximal FE of ~66.1% for ethylamine production with a large partial current density is very low compared to the FE in alkaline condition (Nat. Commun.12(1), 3382 (2021).), and the expensive Pd catalyst compared to Cu catalyst is no superiority. And this work still has not addressed the fundamental issue of hydrogen evolution.

(4) The partial current density, in comparison with the ever-reported HAN electrocatalysts in Figure 2d is unfair because of the different reaction equipment.

(5) Stability test over a span of 20 h using Pd NPs as the cathode catalyst at a constant current of 0.8 A is not enough to demonstrate the stability, can do more time test.

(6) To correlate the acetonitrile hydrogenation activity trends of the studied metal (Ag, Au, Cu, Pt, Pd, W, Mo) catalysts with their intrinsic electronic structures, theoretical analysis should consider the pH value effect. Especially should consider the stability of Cu in acid solution.

To Reviewer #1:

This paper presents an investigation into the electrocatalytic hydrogenation of acetonitrile for ethylamine production in acidic condition using PEM electrolyzers. The authors explore various metallic catalysts and find that Pd/C catalysts exhibit high selectivity (94.4%) and activity (specific production rate of 2762.9 mmol g⁻¹ h⁻¹) at a current density of 200 mA cm⁻², outperforming thermocatalytic processes by two orders of magnitude. While the topic is of interest, there are several concerns and limitations that prevent the paper from being qualified for publication in Nature communications.

Response: We sincerely appreciate the valuable comments raised by the reviewer, which certainly help to improve the manuscript. According to the reviewer's suggestions, we have carefully revised our manuscript and the point-by-point responses are presented below.

1. Comment: One major concern is that the authors propose using proton exchange membrane-based MEA electrolyzers for acetonitrile electroreduction, focusing on acetonitrile electroreduction in acidic conditions. However, the performance of the Pd/C catalysts reported in this paper does not show good performance in PEM-based MEA electrolyzers compared to previous study using Cu catalysts (J. Mater. Chem. A, 2023,11, 2210-2217, Chem Catalysis 1.2 (2021): 393-406).

Response: We thank the referee for the constructive comments and we are pleased to clarify these issues. As for the good performance in the mentioned papers (J. Mater. Chem. A, 2023,11, 2210-2217, Chem Catalysis 1.2 (2021): 393-406), they are actually obtained in different electrolysis condition. The mentioned reference paper used a traditional 3-electrode or flow cell system with alkaline condition, while our work is based on zero-gap PEM-based membrane electrode assembly (MEA) electrolyzers with acidic condition. In addition, the hydrogenation product in acidic condition is ethylamine salt, which is facile for the following product separation due to less species in the solution. In this regard, the studies on E-HAN in acidic media using PEM-MEA might be meaningful. However, the E-HAN in acidic electrolyte is more challenging especially for catalysts development, because the competing hydrogen evolution reaction is more severe in acidic condition, which results in low FE for E-HAN. Aiming at this issue, we have systematically screened various catalysts systems for E-HAN in acidic condition and found Pd displays the highest faradaic efficiency toward ethylamine sulfate. Meanwhile, we also studied the underlying mechanism by various advanced operando characterizations, including Raman, FTIR and XANES spectroscopies. As for the well-developed Cu catalysts in alkaline condition, we found that Cu catalyst has almost no performance in acidic environments due to severe hydrogen evolution side reaction. This also suggests the acidic catalytic system is totally different from the conventional alkaline condition reported in the mentioned reference paper. Since we target different scientific questions in different testing systems, the direct comparison of performances is unreasonable. In order to make the comparison fair, we tested the commonly-used catalysts including W/C, Mo/C, Pt/C, Cu/C, Au/C and Ag/C in the same test conditions, and compared their performance shown in **Figure 2a** and **Figure R1**.

Figure R1. Comparison of the performance of the screened E-HAN catalysts in PEM catalytic reactor.

The authors demonstrate ~70% ethylamine faradaic efficiency (FE) in acidic conditions using Pd/C in an H-type batch cell. However, in a PEM-based MEA electrolyzer, the ethylamine FE drops to 40%, suggesting that pH is not the sole factor determining performance in PEM-based MEA electrolyzers. The microenvironment, including the impact of the PFSA, mass transportation, product crossover, etc., are also important but have not been comprehensively studied.

Response: To unveil the difference of FE between H-type cell and PEM device, we have discussed and studied the microenvironment factors, according to the reviewer's suggestions. 1) First, PFSA generally contributes to the proton conduction in PEMFC and PEMWE in the absence of the acid addition. However, in our work, a high concentration sulfuric acid electrolyte is used, and the proton transfer in the system basically depended on the acidic electrolyte rather than the sulfonate group in the polymer, which should not cause excessive influence on the performance (Nature Energy 6, 475–486 (2021)). 2) We further investigated the acetonitrile and ethylamine crossover from the cathode to the anode of the electrolyzer by testing the NMR data of the products in the anode compartment after 20 h span (**Figure R2**). The NMR results indicates the shuttled ethylamine (~0.02%) and acetonitrile (~0.05%) are negligible within 20 h electrolysis. 3) Considering the mass transportation mainly is related to the local concentration of the solutes, we have investigated the influence of mass transportation by controlling acetonitrile concentration in cathode electrolyte in MEA electrolyzers (**Figure R3**). As acetonitrile concentration increases from 4 wt% to 12 wt%, the ethylamine FE also increases. The ethylamine partial current density achieves 132 mA cm^{-2} at the applied voltage of 0.95 V in 12 wt% acetonitrile, which is 1.11 times as in 8 wt% acetonitrile and 1.18 times as in 4 wt% acetonitrile. It indicates that the mass transfer of acetonitrile in the membrane electrode assembly could be a key factor contributing to the FE difference. We have given brief discussion on it in the revised manuscript.

Additionally, the type of membrane used in this study is not reported, and its stability in the presence of acetonitrile and ethylamine should be investigated. Traditional PEM membranes, such as Nafion, composed of perfluorosulfonic acid, exhibit good stability in water electrolyzers but might not be stable in the presence of organic solvents. Given the strong alkalinity of ethylamine, it is essential to investigate the membrane's stability in extended stability studies.

Response: In this work, Nafion211 with a thickness of $25.4 \mu\text{m}$ was used as the membrane for acetonitrile electroreduction. Considering that organic materials may damage the Nafion membrane, the dissolution of the membrane in acetonitrile and ethylamine solution has also been investigated. A piece of proton exchange membrane with the weight of 22.2 mg is placed in 10 mL 0.5 M H_2SO_4 solution containing 8 wt% acetonitrile and 1 wt % ethylamine for 5 and 10 days, respectively. Due to a large number of F species existing in the membrane, the membrane dissolution is investigated by testing the characteristic signal of dissolved F at -120 ppm in the ^{19}F -NMR spectrum (Phys. Fluids 34, 103113 (2022)). No obvious signal of dissolved F is observed in the solution after placing the membrane in 0.5 M H_2SO_4 solution containing 8 wt% acetonitrile and 1 wt% ethylamine for 5 and 10 days (**Figure R4**), suggesting the membrane is relatively stable in the presence of acetonitrile and ethylamine. We have briefly discussed it in the revised manuscript.

Figure R2. (a) Relationship between ratio of acetonitrile to DMSO in NMR tests and acetonitrile concentrations. (b) NMR data of the exhausted gas of anode.

Figure R3. The FEs of ethylamine and diethylamine, and the corresponding cell voltage versus applied current density on Pd/C testing in 0.5 M H₂SO₄ solution containing 4 wt% acetonitrile (a), 0.5 M H₂SO₄ solution containing 8 wt% acetonitrile (b) and 0.5 M H₂SO₄ solution containing 12 wt% acetonitrile (c). (d) Ethylamine partial current densities under different concentrations of acetonitrile.

Figure R4. ^{19}F NMR spectra of the membrane placed in 0.5 M H_2SO_4 containing 8 wt% acetonitrile and 1 wt% ethylamine for 5 and 10 days. The highlighted region is the location of characteristic signal of Nafion.

Furthermore, considering the low faradaic efficiency of acetonitrile in acidic conditions, it might be more sensible to use an anion exchange membrane-based MEA electrolyzer capable of running acetonitrile electroreduction in neutral or alkaline conditions.

Response: Although there have been significant progresses in alkaline and neutral E-HAN in literature, the acidic E-HAN has its own advantages on product separation and the maturity of PEM membrane yet is rarely studied. Aiming at the low FE of E-HAN in acidic condition stemming from the severe hydrogen evolution side reaction, we herein systematically screened various catalysts systems for E-HAN in acidic condition and found Pd displays the highest faradaic efficiency toward ethylamine sulfate. Meanwhile, we also studied the underlying mechanism by operando characterizations such as Raman, FTIR and XANES spectroscopies and DFT calculations. Studying the performance in AEM cell will deviate from the targeted scientific questions of this work, and could make confused data presentation. As for the E-HAN using AEM membrane, it has been studied in our previous work by using conventional copper catalysts (Nat. Commun. 14, 3847 (2023)). However, the conventional copper catalyst is not suitable for acidic E-HAN catalysis due to its severe side reactions, which has been demonstrated in this work.

The authors should provide a clearer explanation of how their work extends the existing knowledge and contributes to the field.

Response: In this work, we target to address the low FE of E-HAN in acidic condition stemming from the severe hydrogen evolution side reaction. Although alkaline and neutral condition have been well developed, acidic E-HAN catalysis is rarely studied, because it raises great challenges on catalysts. Considering that no effective catalysts have been reported in acidic environments, we systematically screened various catalysts systems for E-HAN in acidic condition and found Pd displays the highest faradic efficiency toward ethylamine sulfate. The reaction mechanism is further investigated by various advanced operando characterizations, including Raman, FTIR and XANES spectroscopies, which unveil that the in-situ formed PdH_x is the active centers for catalytic reaction and the adsorption strength of the $^*\text{MeCH}_2\text{NH}_2$ intermediate determines the catalytic selectivity. Moreover, the theoretical analysis reveals a classic d-band mediated volcano curve to describe the relation between the electronic structures of catalysts and activity, providing valuable insights for designing acidic E-HAN catalysts.

2. Comment: Although the authors report a 94% selectivity towards ethylamine, the faradaic efficiency is very low (<40%) at a current density of 200 mA cm^{-2} , compared to previous reports (>90%), indicating inefficient utilization of electrons.

Response: Thank you for the valuable comments. As we have discussed above, due to the targeted scientific questions and testing conditions are different between our work and the

previous reports, the direct comparison of performances is unfair. Actually, Pd is the best performance among the commonly-used catalysts in acidic system. The main target of this work is to screen potential catalyst for acidic acetonitrile reduction for PEM electrolyzer and reveal the underlying structure-property relation in the acidic acetonitrile reduction, which could provide valuable reference for developing highly efficient E-HAN catalysts in the future.

Moreover, performing hydrogen oxidation at the anode to provide protons decreases the cell voltage but compromises the advantage of using water as a proton source in electrochemical hydrogenation. Considering the low faradaic efficiency of acetonitrile electroreduction reported in this paper, where 60% of hydrogen consumed at the anode does not contribute to ethylamine production, there is a 1.5 times higher hydrogen consumption compared to traditional thermocatalytic acetonitrile hydrogenation using hydrogen as a proton source.

Response: Although using water as a proton source in electrochemical hydrogenation is one of the advantages in PEM electrolyzer, the high voltage (> 1.23 V) and precious IrO_2 catalyst for oxygen evolution reaction in the acidic condition unavoidably increase the capital cost. Using hydrogen oxidation in the anode, instead, requires low driven voltage (> 0 V) and less expensive Pt/C catalyst, which would decrease the total cost of the cell. Besides, although the apparent usage of hydrogen in the PEM electrolyzer is almost 1.5 times more than that in thermocatalytic system, the gas phase hydrogen generated at the cathode can be easily separated from the liquid product and recycled for the reuse at the anode, which could solve the mentioned the issues.

Additionally, it is confusing to report selectivity and faradaic efficiency separately. I would suggest that the authors clarify that selectivity refers to selectivity towards primary amine and report the faradaic efficiency simultaneously.

Response: To avoid the confusion on selectivity and faraday efficiency, as suggested, we use faradaic efficiency instead of selectivity in the revised manuscript.

3. Comment: The authors suggest in the SR-FTIR study that the vibration band intensities of CCN and CH_3 for ethylamine synthesis on Cu, Pt, and Au surfaces are relatively lower than those on Pd, indicating that MeCN molecules are less likely to be adsorbed and activated on Cu, Pt, and Au surfaces compared to Pd. However, SR-FTIR cannot quantitatively compare adsorption energy across different samples, so drawing conclusions based solely on band intensities is not appropriate. To accurately assess the adsorption energy differences on various catalysts, direct experimental characterization methods should be employed.

Response: We appreciate the raised valuable comments. We agree that SR-FTIR cannot quantitatively compare adsorption energy across different samples. Since SR-FTIR can give information about the transformation behavior of acetonitrile, we have revised our description on infrared characterization and emphasized that SR-FTIR were performed to study the acetonitrile conversion process on different catalysts. Compared with the impressive band intensity change at different potentials on Pd catalyst surface, the vibration band intensities of CCN, CH_3 for ethylamine synthesis on Cu, Pt, and Au are relatively lower and change insignificantly at different potentials, indicating that the adsorption and transformation of MeCN molecules on Cu, Pt, and Au surfaces are quite limited. To directly analyze the adsorption strength of the intermediate molecules, temperature programmed desorption (TPD) could be a useful method. We have discussed with the technician of TPD that it is not feasible for TPD to detect liquid molecules as their instruments cannot produce a stable gas flow from acetonitrile. Alternatively, the open-circuit potential (OCP), reflecting adsorbates in the inner Helmholtz layer, is another characterization method to assess the adsorption behavior over different catalysts (Angew. Chem. Int. Ed. 2023, e202300094; ACS Nano 2022, 16, 21518–21526). In general, after adding organic material, the larger decrease of OCP indicates stronger adsorption of the organic molecules on catalyst surface. Upon the addition of 8 wt% MeCN, a more significant decrease in OCP for Pt/C (0.1338 V) and Pd/C (0.0639 V) was observed than that of Cu/C ($\Delta = 0.009$ V), Au/C ($\Delta = -0.0006$ V) and Ag/C ($\Delta = 0.0346$ V), suggesting stronger acetonitrile adsorption on Pt and Pd surfaces (**Figure R5**). This trend is

basically consistent with our theoretical calculation results (Figure 5). We have briefly discussed it in the revised manuscript.

Figure R5. OCP of Pt/C (a), Pd/C (b), Cu/C (c), Au/C (d) and Ag/C (e) in 0.5 M H₂SO₄ solution before and after 8% MeCN was added.

4. Comment: The paper compares the activity of acetonitrile electroreduction with conventional thermocatalytic acetonitrile hydrogenation and claims a production rate of 2762.9 mmol g⁻¹ h⁻¹ using Pd/C catalysts, outperforming thermocatalytic processes by two orders of magnitude. However, comparing mass activity between thermocatalysis and electrocatalysis is unfair, considering the small amount of electrocatalysts used. It would be more comprehensive to evaluate the performance of Pd/C catalysts by comparing them with other electrocatalysts only.

Response: Thank you for your valuable suggestion. We agree that direct comparison between thermocatalysis and electrocatalysis is unfair. Following the suggestions, we only compare the performance of electrocatalysts. As discussed in the previous response, the acidic catalytic system is totally different from the conventional alkaline condition, which make the direct comparison is also unreasonable. In order to make the comparison fair, we tested the commonly-used catalysts including Pd/C, Cu/C, W/C, Mo/C, Pt/C, Au/C and Ag/C in the same test conditions, and compared their performance shown in Figure R1. The result shows that the PEM catalytic reactor with Pd/C catalyst achieves a specific activity of 2912.5 mmol g⁻¹ h⁻¹ at a relatively low voltage, which is almost an order of magnitude higher than the rest electrocatalysts, verifying the superior advancement of the E-HAN catalyzed by the Pd/C.

5. Comment: I recommend that the authors report the current density instead of the total current in Figure 3 and specify the active area of the electrode.

Response: We sincerely appreciate the reviewer's insightful suggestion. We have replaced the total current in Figure 2 by the current density, and specify the active area of the electrode in the experimental section in the revised manuscript.

Overall, while the topic of electrochemical hydrogenation of acetonitrile for ethylamine production is of interest, the paper does not meet the standards required for publication in Nature communications. The authors should address the mentioned concerns, provide additional investigation on the microenvironmental impact in the PEM-based MEA electrolyzer, clarify the stability of the membrane in organic solvents, refine the comparisons made, and improve the reporting of experimental data to strengthen the manuscript.

Response: We sincerely thank the referee for carefully reviewing our manuscript and recognizing the topic of our study is interesting. We also appreciate the valuable comments

which certainly help to improve our manuscript. According to the reviewer's suggestions, we have prepared a point-by-point response to address the raised concerns.

To Reviewer #2:

Wang, Xiao, Duan and colleagues reported the electrochemical hydrogenation of acetonitrile (EHAN) to ethylamine using a PEM electrolyzer. PEM electrolyzer features zero-gap configuration of electrodes and no added-electrolyte in solution, thus holding a promise in the industrial electrochemical production of chemicals. While a recent report by Jiao and co-workers revealed that EHAN in alkaline media using Cu catalyst is a promising technology, the present work featured some advantages such as small ohmic loss and larger current density. This work seems to propose a promising technology to electrify the ethylamine industry. However, some serious scientific concerns listed below should be addressed prior to further consideration.

Response: We thank the reviewer for the positive comment to our work. We also appreciate the valuable comments raised by the reviewer, which help us improve the manuscript. According to the reviewer's suggestions, we have carefully revised our manuscript.

1. Comment: The catalyst screening is problematic. Firstly, the preparation method for Cu/C, W/C, and Mo/C largely differs from that of Pt/C, Au/C and Ag/C. More precisely, Pt/C, Au/C and Ag/C were prepared by chemical reduction of corresponding salts in the presence of carbon support, while Cu/C, W/C, and Mo/C were prepared by physically mixing carbon and metal powder. Thus, the nature of these catalysts is considerably different. Hence, the reviewer does not believe that it is scientifically not appropriate to compare the reactivity of this catalyst. Furthermore, it is misleading to discuss experimental catalytic activity and computational results in parallel, since the nature of catalysts is, again, very different.

Response: We appreciate the professional and useful comments. To keep the same preparation method for these catalysts, we have changed the synthesis methods of Pt/C, Ag/C, and Au/C. Specifically, we first synthesized their nanoparticles through the tannic acid protection method, and then mixing the nanoparticles with carbon (XC-72) to synthesize Pt/C, Ag/C and Au/C (experimental methods, page 14). In order to eliminate the difference in activity caused by different nature of catalysts, we tested the electrochemical surface areas (ECSA) of these nanoparticles (without carbon loading) and normalized the activity to the ECSA (**Figures R1-2**). It can be seen from the **Figures R3-R4** that the ECSA normalized activity trend is basically the same as before, and Pd still has the best specific performance.

Figure R1-2 has been added in supplementary information as Supplementary Figures 10-11 and corresponding discussion has been added in the revised manuscript.

Figure R1. The cyclic voltammetry curves of W (a), Mo (b), Pt (c), Pd (d), Cu (e), Au (f) and Ag (g) nanoparticles, respectively. The tests are performed under Ar atmosphere at different scan rates.

Figure R2. The plots of ΔJ versus scan rates for W (a), Mo (b), Pt (c), Pd (d), Cu (e), Au (f) and Ag (g) nanoparticles, respectively.

Figure R3. (a) Potential-dependent ethylamine partial current density on various metal catalysts. (b) Ethylamine partial current densities normalized by ECSA on various metal catalysts.

Figure R4. (a) Mass activity profile versus both ΔG_{MeCN} and $\Delta G_{\text{MeCH}_2\text{NH}_2}$. (b) Ethylamine partial current densities normalized by ECSA (J_{ECSA}) profile versus both ΔG_{MeCN} and $\Delta G_{\text{MeCH}_2\text{NH}_2}$.

2. Comment: For the catalyst screening, the authors should disclose V-t curve for the experiment. The reviewer is wondering if it is possible to use non-precious metals such as Cu in acidic media. If metal catalysts are dissolving, the potential plausibly dramatically changes during the electrolysis. If a non-precious metal catalyst does not survive in this media, then further discussion on the catalyst activity (such as computations) should be avoided.

Response: We thank the constructive comments and we are pleased to clarify this issue. As suggested, we have disclosed the V-t curves for PEM-based MEA tests (**Figure R5**). As it can be seen from **Figure R5**, these catalysts can work well in MEA electrolyzer for a short-term test. In order to study the metal stability in acidic media more comprehensively, the J -t curves for the screened catalysts using three-electrode system are also displayed (**Figure R6** and **Supplementary Figure 8**). According to the order of the metal activity table, since the metal activity of Cu is more inert than H, Cu is relatively stable in acidic media. Moreover, Cu is also used in other acidic catalytic system such as CO₂ reduction in literature (Science 372, 1074–1078 (2021)). Therefore, Cu is relatively stable for a short-term test under the reduction potential in the acidic environment. For long-term stability of copper catalysts, it is beyond the scope of this work. We have added a brief discussion on it in the revised manuscript.

Figure R5. V-t curves for MEA tests of W/C (a), Mo/C (b), Pt/C (c), Pd/C (d), Cu/C (e), Au/C (f) and Ag/C (g), respectively.

Figure R6. J-t curves for three electrode tests of W/C (a), Mo/C (b), Pt/C (c), Cu/C (d), Au/C (e) and Ag/C (f), respectively.

3. Comment: The author did not find NMR data for catalyst screening. Also, TEM is not shown for catalysts other than Pd/C. These data should be disclosed.

Response: We appreciate the valuable comments. As suggested, we have disclosed the NMR data and the TEM images for the screened catalyst in the revised supplementary information (Figures R7-8).

Figure R7. The ^1H NMR spectra of electrolyte products by Cu/C at -0.29 V (a), Pt/C at -0.24 V (b), Au/C at -0.324 V (c), Ag/C at -0.29 V (d), W/C at -0.383 V (e) and Mo/C at -0.357 V (f).

Figure R8. TEM images of W/C (a), Mo/C (b), Pt/C (c), Cu/C (d), Au/C (e) and Ag/C (f), respectively.

4. Comment: There is no information on the conversion and the yield of EHAN process presented herein. These aspects should also be discussed for a fair comparison with preceding systems.

Response: We appreciate the constructive suggestion. In electrocatalysis process, the liquid flow rate in the membrane electrode chamber is rapid, resulting in a short reaction time and a low single-pass conversion (1.35%), which could be improved by recycling the electrolyte solution. However, from the perspective of yield, Pd catalyst in the MEA electrocatalysis system is two orders of magnitude higher than the thermocatalytic process. Since the reaction conditions of electrocatalysis and thermal catalysis are completely different, and the direct comparison is not reasonable. Aiming at this issue, we have added a brief discussion on it in the revised manuscript. Considering the small amount of electrocatalysts used currently, we herein compare the catalytic performance of several screened catalysts (Pt/C, Au/C, Mo/C, Ag/C and W/C) in PEM catalytic electrolyzer (**Figure R9**). Meanwhile, for better comparison,

the performance of the three-electrode system in our works is compared with the most relevant electrocatalyst of acetonitrile hydrogenation in the literature (Table R1).

Figure R9. Comparison of the performance of the screened E-HAN catalysts in PEM based MEA electrolyzer.

Table R1. Comparison of the performance of the Pd NPs in acidic condition with previous E-HAN systems.

Catalysts	Test condition	Patial current density at 0.5 V vs. RHE (mA cm ⁻²)	Faradaic efficiency at 0.5 V vs. RHE	References
Pd NPs	8 wt% MeCN + 0.5 M H ₂ SO ₄	212	60.7%	This work
Cu NSs	0.5 M MeCN + 0.5 M KHCO ₃ in CO ₂ atmosphere	9	96%	Chem Catalysis 1, 393-406 (2021)
	0.5 M MeCN + 0.5 M KHCO ₃ in Ar atmosphere	3.2	40%	
Cu ₃ Ni ₁ MAs	1 M NaOH with 8 wt% MeCN	9.6	90%	J. Mater. Chem. A, 11, 2210-2217 (2023)
Cu		107	91%	
Pd	1 M NaOH with 8 wt% MeCN	74	68%	Nat. Commun. 12, 1-8 (2021)
Ni		52	53%	

5. Comment: (1) The present system uses hydrogen oxidation reaction as an anodic reaction. The reviewer believes the advantage of PEM cell is the use of water as a source of protons and electrons. If hydrogen gas is used, then the process requires H₂ tank, making the process much more complicated. (2) Also, H₂ gas is currently produced by steam reforming of methane, which causes an increase in CO₂ emission. (3) Also, the current efficiency is limited to 35% for 20 h electrolysis as shown in Figure 3e, which means 65% of H₂ gas is basically wasted. The authors should make some comments on this point, and discuss if this process can still compete with the thermal process.

Response: We appreciate the insightful comments raised by the reviewer. (1) The catalytic reaction using water as the proton source in the PEM cell typically requires the use of very

expensive IrO₂ catalysts in the anode, while using H₂ gas as the proton source only requires relatively costless Pt/C. In addition, the thermodynamic potential of the HOR reaction is only 0 V, while the OER requires 1.23 V. The tandem of HOR and HAN only requires low energy input. Besides, similar to hydrogen-oxygen fuel cell technology, the HOR reaction occurring at the anode is already a very mature technology.

(2) With development of clean energies, such as hydropower, wind power and photovoltaic technology, the cost of green hydrogen will be significantly lowered in the future, and the ratio of hydrogen produced by methanol reforming and the corresponding CO₂ emission would be reduced.

(3) Although the apparent usage of hydrogen in the PEM electrolyzer is almost 1.5 times that in thermocatalytic system, the gas phase hydrogen generated at the cathode can be easily separated from the liquid product and recycled for the reuse at the anode, which could economize the amount of hydrogen used in the anode. PEM cell can be made into large stacks, enabling mass production at miniaturized equipment and saving land costs. Besides, the PEM electrolyzer can be performed at normal temperature and pressure, which solves the concerns on safety and economic costs of thermocatalytic process. We have added a brief discussion on it in the revised manuscript.

6. Comment: The authors should do some calculations on how much it cost to produce ethylamine using this process.

Response: We thank the referee for the valuable comment. Based on the suggestions of the referee, techno-economic analysis (TEA) of electrocatalytic acetonitrile hydrogenation to ethylamine has been investigated. Specifically, preliminary techno-economic analysis (TEA) estimates the net revenues of ~\$907.66 for upcycling per ton of acetonitrile under industrial-level current density (~200 mA cm⁻²), as shown in **Figure R10**. We have incorporated the suggestions made by the reviewer into the revised manuscript and have added corresponding discussions. Additionally, **Figure R10** and **calculation detail** have been included in the **Supplementary Information**.

Figure R10. Techno-economic analysis (TEA) of electrocatalytic acetonitrile hydrogenation to ethylamine at 200 mA cm⁻².

7. Comment: Operando IR in PEM cell using Pd/C catalyst is recently reported by Kondo and Atobe, where on top Pd-H species was successfully observed for the first time (DOI: 10.1021/acs.jpcc.2c05127). It would be informative to discuss such on top Pd-H species in this system.

Response: We thank the referee for providing the valuable reference. Kondo and Atobe first reported the operando IR spectra of the top hydrogen on the Pd cathode (Pd-H) at 2030 cm⁻¹ in

PEM cell. The method developed by the authors enables direct observation of adsorbed species on metal catalysts in PEM reactors, is of huge potential in the development of catalysts for a wide range of applications in electrocatalysis. This article deepens our understanding of the evolution of Pd during in situ processes. We have now referred this article in our discussion on *in situ* SR-FTIR characterization of Pd-H species (**reference 33**).

8. Comment: The use of PEM electrolyzer for the production of fine chemicals is gaining a high level of attention. Especially, the group of Atobe and Shida demonstrated a variety of reactions using PEM electrolyzer. Their manuscripts should be properly cited. Representative Articles: DOI: 10.1021/acseenergylett.2c02573 DOI: 10.1021/acscatal.2c01594 DOI: 10.1021/acssuschemeng.9b01882

Response: Thank you for the valuable suggestion. We have carefully read the literature you mentioned (DOI: 10.1021/acseenergylett.2c02573, DOI: 10.1021/acscatal.2c01594, DOI: 10.1021/acssuschemeng.9b01882). These literatures have reported the demonstration of multiple reaction systems (hydrogenation of Cyclic Ketones, hydrogenation of Alkyne and semihydrogenation of Alkynes) in PEM electrolyzers, which are relevant to our research. As suggested, we have referred the relevant reference in the revised manuscript (**reference 19, 20 and 24**).

To Reviewer #3:

General comment:

Electrochemical hydrogenation of acetonitrile based on acid proton exchange membrane electrolyzers still faces huge challenge because the local acidic condition of PEM results in severe competitive proton reduction reaction and poor selection toward E-HAN. Here, the authors conduct a systematic study to screen various metallic catalysts and discover Pd/C catalysts exhibits good ethylamine selectivity at the high current density, the results are very interesting, However, the research system lacks innovation for Nat. Commun, and there are still some problems to be solved:

Response: We sincerely thank the referee for carefully reviewing our manuscript and recognizing the results are interesting. As for the novelty, we are pleased to clarify this issue. Although there have been significant progresses in alkaline and neutral E-HAN in literature, the acidic E-HAN has its own advantages on product separation yet is rarely studied. However, the E-HAN in acidic electrolyte is more challenging especially for catalysts development, because the competing hydrogen evolution reaction is more severe in acidic condition, which results in low FE for E-HAN. Since no effective catalysts have been reported in acidic environments, we have systematically screened various catalysts systems for E-HAN in acidic condition and found Pd displays the highest faradaic efficiency. Moreover, our spectroscopy studies unveil that in-situ formed PdH_x is the active centers for catalytic reaction and the adsorption strength of the *MeCH₂NH₂ intermediate determines the catalytic selectivity. Meanwhile, theoretical analysis reveals a classic d-band mediated volcano curve to describe the relation between the electronic structures of catalysts and activity, providing valuable insights for designing acidic E-HAN catalysts. In comparison with the conventional alkaline and neutral system, we target a quite different scientific questions and obtain new understanding on catalyst design. Therefore, we reasonably believe the novelty of our work is suitable for *Nature Communications*.

1. Comment: The different catalysts is not easy to compared due to some structures difference of catalysts, like the size, morphology, active surface area etc., the comparison of the activity of different catalysts is unscientific.

Response: We thank the referee for the valuable comment. In order to eliminate the difference in activity caused by different nature of catalysts, we tested the electrochemical surface area (ECSA) of these nanoparticles (without carbon loaded) and normalized the activity to the ECSA (Figures R1-2). It can be seen from the Figure R3 that the ECSA normalized activity trend is basically the same as before, and Pd still has far leading specific performance. We believe that the activity trend after normalization has reference value for understanding the catalytic activity.

Figure R1. The cyclic voltammetry curves under Ar atmosphere at different scan rates of W (a), Mo (b), Pt (c), Pd (d), Cu (e), Au (f) and Ag (g) nanoparticles, respectively.

Figure R2. The plots of ΔJ versus scan rates for W (a), Mo (b), Pt (c), Pd (d), Cu (e), Au (f) and Ag (g) nanoparticles, respectively.

Figure R3. (a) Potential-dependent ethylamine partial current density on various metal catalysts. (b) Ethylamine partial current densities normalized by ECSA on various metal catalysts.

2. Comment: The ethylamine selectivity of 94.4% is easy to confused because that doesn't contain hydrogen evolution.

Response: Thank you for the valuable suggestion. To eliminate unnecessary misunderstandings, we have now changed the ethylamine selectivity to faraday efficiency (FE) in the revised manuscript.

3. Comment: The Pd nanoparticles display a maximal FE of ~66.1% for ethylamine production with a large partial current density is very low compared to the FE in alkaline condition (Nat. Commun.12(1), 3382 (2021)), and the expensive Pd catalyst compared to Cu catalyst is no superiority. And this work still have not address the fundamental issue of hydrogen evolution.

Response: We thank the referee for the valuable comments, and we are glad to clarify this issue. In this work, we target to address the low FE of E-HAN in acidic condition stemming from the severe hydrogen evolution side reaction. Although alkaline and neutral conditions have been well developed, acidic E-HAN catalysis is rarely studied, because it raises great challenges on catalysts. Considering that no effective catalysts have been reported in acidic environments, we systematically screened various catalysts systems for E-HAN in acidic condition and found Pd displays the highest faradic efficiency. As for the well-developed Cu catalysts in alkaline condition, we found that Cu catalyst has almost no performance in acidic environments due to severe hydrogen evolution side reaction. This also suggests the acidic catalytic system is totally different from the conventional alkaline system as reported in the mentioned reference paper. Since we target different scientific questions in different testing systems, the direct comparison of performances is unreasonable. Moreover, Pd displays the best performance with minimal hydrogen evolution activity among the screened catalysts in acidic system. In this work, the main target is to search potential catalyst for acidic acetonitrile reduction for PEM electrolyzer and reveal the underlying structure-property relation. We believe the cost and activity of catalysts could be further optimized by surface engineering in future.

4. Comment: The partial current density, in comparison with the ever-reported HAN electrocatalysts in Figure 2d is unfair because the different reaction equipment.

Response: We thank the referee for the valuable suggestion. Following the suggestion, we have removed the comparison with the ever-reported HAN electrocatalysts in the revised manuscript (Figure 2). For a fair comparison, we compared the performance of different catalysts in H-type cell and MEA electrolyzer, respectively, with same test conditions (Figure R3 and R4), which is included in the revised manuscript and Supplementary Information.

Figure R4. Comparison of the performance of the screened E-HAN catalysts in PEM based MEA electrolyzer.

5. Comment: Stability test over a span of 20 h using Pd NPs as the cathode catalyst at a constant current of 0.8 A is not enough to demonstrate the stability, can do more time test.

Response: Thank you for the valuable comment. As suggested, we have extended stability test time to 45 h using Pd NPs as the cathode catalyst at a constant current of 0.8 A (**Figure R5**). The cell maintains a stable cell voltage (~0.89 V) and a steady ethylamine FE (>35%) for 45 h, demonstrating its reasonable stability.

Figure R5. Stability test over a span of 45 h using Pd NPs as the cathode catalyst at a constant current of 0.8 A.

6. Comment: To correlate the acetonitrile hydrogenation activity trends of the studied metal (Ag, Au, Cu, Pt, Pd, W, Mo) catalysts with their intrinsic electronic structures, theoretical analysis should consider the pH value effect. (2) Especially should consider the stability of Cu in acid solution.

Response: We appreciate the reviewer's constructive comments. (1) As for the effects of pH in calculation, the chemical potential for the reaction ($H^+ + e^-$) is equal to that of $1/2H_2 - k_B T \ln 10 \times \text{pH}$, when setting the reference potential to be the standard hydrogen electrode (SHE) potential with different pH (J. Phys. Chem. C 112, 9872–9879 (2008)). Since 0.5 M H_2SO_4 electrolyte was used in our test system and the pH related term is 0, it is difficult to distinguish the effect of pH on performance from the perspective of calculation method.

However, in the actual acetonitrile reduction test system, the acidity and alkalinity play vital effects. Considering the hydrated proton is kinetically favorable for reduction in acid than that in base, Cu catalysts that perform well under alkaline condition undergo severe hydrogen evolution side reaction in acidic condition. In order to fully understand the activity trend of different metals under acidic conditions, the adsorption energy of hydrogen (ΔG_{H^*}) should also be considered, which is related to the activity of side reaction. The large $|\Delta G_{H^*}|$ and small ΔG_{RDS} indicate that catalyst surface is favorable for acetonitrile hydrogenation but not for the hydrogen evolution reaction. Thus, $|\Delta G_{H^*}| - \Delta G_{RDS}$ can be used as the other descriptive factor. With $|\Delta G_{H^*}| - \Delta G_{RDS}$ as the horizontal coordinate and ΔG_{RDS} as the vertical coordinate, it can be determined

that the catalyst show better performance as its coordinate is close to the lower right (**Figure R6**). Among these metals, Pd-based catalysts have the best catalytic performance with various *H coverages, which is significantly higher than that of Cu. In summary, considering the different proton sources in acid and base, acetonitrile hydrogenation activity trends should be different.

Figure R6. Relationship between $|\Delta G_{H^*}| - \Delta G_{RDS}$ and ΔG_{RDS} at *H coverage of 0 ML (a) and under various *H coverages (b).

(2) It is basically difficult to consider the stability of Cu by theoretical calculation, because the calculated time scale (ps) is inconsistent with the real test time scale (s). To consider the stability of copper in acidic condition, we have experimentally tested the J - t curve in three electrode system of Cu/C catalysts and the V - t curve in MEA system and found that Cu is relatively stable for a short-term test in acidic solution (**Figure R7**). According to the order of the metal activity table, since the metal activity of Cu is more inert than H, Cu is relatively stable in acidic media, which is also used in other acidic catalytic system such as CO₂ reduction in literature (Science 372, 1074–1078 (2021)). Therefore, Cu is relatively stable for short-term studies under the reduction potential in the acidic environment. For long-term stability of copper catalysts, it is beyond the scope of this work. We have added a brief discussion on it in the revised manuscript.

Figure R7. (a) J - t curves of Cu/C in three-electrode tests. (b) V - t curves of Cu/C in MEA tests.

REVIEWER COMMENTS

Reviewer #1 (Remarks to the Author):

I have carefully reviewed the revised manuscript and the authors' responses to my previous comments. Unfortunately, several critical issues remain inadequately addressed, which leads me to recommend rejection of this manuscript in its current form. Below, I detail the main concerns that require significant attention:

1. The manuscript does not convincingly justify the importance of investigating acetonitrile electrochemical hydrogenation in acidic conditions. Given the extensive studies in neutral/alkaline conditions, the modest 40% Faradaic efficiency (FE) towards ethylamine in acidic conditions seems unremarkable. Moreover, the formation of ethylamine sulfate in acidic conditions raises questions about its industrial applicability compared to ethylamine. Clarification on whether ethylamine sulfate has similar applications or feasible conversion methods back to ethylamine is necessary.
2. The discussion on acetonitrile electroreduction in PEM electrolyzers is insufficient, particularly regarding the claimed 40% FE towards ethylamine compared to 70% in H-cell. The manuscript oversimplifies the differences between the microenvironment on PFSA membranes and sulfuric acid, neglecting significant aspects like the reaction kinetics at the electrochemical interface. Additionally, the assertion that mass transport is the only differing factor between H cells and PEM electrolyzers is unconvincing, given the expected superior mass transport in flow electrolyzers like PEM. The PEM electrolyzer suppose to have better mass transport of reactant to the catalysts which is favorable for acetonitrile electroreduction. It is not convincing to attribute the low FE in PEM electrolyzer to the mass transport limitation.
3. The investigation into the stability of the Nafion membrane is incomplete. The focus should not only be on membrane dissolution in acetonitrile but also on critical factors like swelling and loss of ion exchange capacity. These aspects are best evaluated under operational conditions of the electrolyzer, not merely by dissolving the membrane in a solution.
4. The manuscript lacks a comprehensive economic feasibility comparison of hydrogen oxidation at the anode versus oxygen evolution reaction. The argument that using Pt/C could lower the total costs compared to IrO₂, especially considering the consumption of hydrogen over water, is not substantiated with adequate economic analysis.
5. The interpretation of SR-FTIR data, particularly the vibration band intensities of CCN and CH₃ of ethylamine, is not accurate and insightful. The strong signals corresponding to ethylamine on Pd, compared to Cu, Pt, and Au, are expected due to Pd's efficacy in acidic acetonitrile electroreduction. This observation does not provide mechanistic insights into why Pd is the preferred catalyst for acetonitrile reduction in acid.

In summary, while the manuscript addresses a topic of potential interest, these significant gaps in rationale, experimental design, and data interpretation prevent me from recommending its publication in its current state.

Reviewer #2 (Remarks to the Author):

The authors have answered all questions raised by the reviewers. The reviewers appreciate the authors' dedication to this revision effort. Most additional data and responses were adequate.

There are still potential challenges to overcome in the proposed E-HAN process, notably the very low conversion of starting materials (only 1.35% in the single-pass experiment). The reviewers believe that this value is too low to imagine real-world applications.

Even so, this paper demonstrates the high potential of the PEM system as an E-HAN process, which gives important insights into electrifying the current energy-consuming thermal processes.

After the revision, this manuscript is in good shape and meets the high criteria and broad interest of Nature Communications. Thus, the reviewer recommends the acceptance of the manuscript for publication.

Reviewer #3 (Remarks to the Author):

The authors have modified the manuscript according to the reviewer's advice, but I still think this study is very reluctant due to the low FE (43.8%) in the acidic condition of PEM and use the noble metal catalysts compared to the Cu catalyst in alkaline condition, and from the revised supporting information in Figure R10, we can know that the economy is not very good although the authors think the acidic E-HAN has its own advantages on product separation.

Response to Reviewers

We sincerely appreciate the reviewers for spending their valuable time evaluating our manuscript and giving us constructive suggestions to improve the quality of the manuscript. The point-by-point responses to the reviewers' comments are attached below and all the corresponding revisions newly made are highlighted using green in the revised manuscript.

To Reviewer #1:

I have carefully reviewed the revised manuscript and the authors' responses to my previous comments. Unfortunately, several critical issues remain inadequately addressed, which leads me to recommend rejection of this manuscript in its current form. Below, I detail the main concerns that require significant attention:

Response: Thank you for your insightful comments and valuable suggestions that help us improve the quality of the manuscript. We have followed your comments and substantially revised our manuscript.

1. The manuscript does not convincingly justify the importance of investigating acetonitrile electrochemical hydrogenation in acidic conditions. Given the extensive studies in neutral/alkaline conditions, the modest 40% Faradaic efficiency (FE) towards ethylamine in acidic conditions seems unremarkable.

Response: We deeply thank the referee for the constructive comments and we are pleased to clarify these concerns. The acidic acetonitrile reduction generally benefits from the more mature development of proton exchange membrane and is free of electrolyte pollution induced by CO₂ in air, a common concern in alkaline systems (Nat. Catal., 2022, 5, 268–276; Nat. Commun., 2023, 14, 8036).

Although there exist some important studies on acetonitrile reduction in neutral and alkaline media, the main focuses and innovations of this manuscript are quite different. In detail, this manuscript firstly attempts to apply acetonitrile reduction in the well-developed PEM electrolyzer. Meanwhile, to address the hydrogen evolution side reaction in acidic media, we have screened several potential metallic catalysts for the reaction and we have achieved the best performance using Pd catalyst. More importantly, we have established the linkage between the acidic acetonitrile reduction activity and the adsorption strength of the catalysts based on various operando characterizations and theoretical calculations, which could help to further design more efficient catalysts.

For the Faradaic efficiency, the ~40% ethylamine Faradaic efficiency is obtained in PEM electrolyzer, while the majority of the reported high Faradaic efficiencies in alkaline or neutral media were obtained in three-electrode system. The cross comparison between different types of electrolyzers is not fair. In our study, the Faradaic efficiency reaches to ~67% in three-electrode system, indicating the intrinsic activity of Pd towards acidic acetonitrile reduction is also high. We do admit the Faradaic efficiency in our study is lower than previously reported catalyst in alkaline and neutral conditions. However, in our study, for better screening, all the catalysts are relatively "raw" materials without special modifications. Thus, it is unfair to compare

the catalyst for the acidic PEM electrolyzer in our studied with the carefully modified catalysts for alkaline or neutral electrolyzer. In addition, similar to the acidic CO₂ reduction, although the Faradaic efficiency of CO₂ reduction in acid media was lower than in alkaline or neutral media, a significant development has been witnessed in recent years, and acidic CO₂ reduction is currently an important topic in electrolysis.

Thus, in this manuscript, we are aiming at putting forward a new strategy and presenting some important insights into the catalyst designing for acetonitrile hydrogenation, which could further provide guidance for designing acidic acetonitrile reduction catalysts for the well-developed PEM electrolyzer in the future.

Moreover, the formation of ethylamine sulfate in acidic conditions raises questions about its industrial applicability compared to ethylamine. Clarification on whether ethylamine sulfate has similar applications or feasible conversion methods back to ethylamine is necessary.

Response: Ethylamine sulfate, as one of ethylamine salt, has been being used in the organic synthesis (*Angew. Chem. Int. Ed.*, 2015, 54, 3768–3772; *J. Phys. Chem. B*, 2006, 110, 22479–22487) and pharmaceuticals (*Pharm. Technol.*, 2008, 32, 128–146; *Pharm. Technol.*, 2007, 31, 78–84), and it has also been used as reaction media (*RSC Adv.*, 2015, 5, 71449–71461; *J. Phys. Chem. B*, 2017, 121, 4592–4599), which could demonstrate its applications.

For the conversion of ethylamine sulfate into ethylamine, it can be converted to ethylamine by a simple metathesis reaction, where the ethylamine could be liberated by strong base such as KOH (*Sep. Purif. Technol.*, 2021, 277, 118229) with the production of the value-added K₂SO₄ by-product.

2. The discussion on acetonitrile electroreduction in PEM electrolyzers is insufficient, particularly regarding the claimed 40% FE towards ethylamine compared to 70% in H-cell. The manuscript oversimplifies the differences between the microenvironment on PFSA membranes and sulfuric acid, neglecting significant aspects like the reaction kinetics at the electrochemical interface. Additionally, the assertion that mass transport is the only differing factor between H cells and PEM electrolyzers is unconvincing, given the expected superior mass transport in flow electrolyzers like PEM. The PEM electrolyzer suppose to have better mass transport of reactant to the catalysts which is favorable for acetonitrile electroreduction. It is not convincing to attribute the low FE in PEM electrolyzer to the mass transport limitation.

Response: We thank the referee for the valuable comments, and we are glad to clarify these issues. The phenomenon that the FE in membrane electrode assemblies (MEA) is lower than in H-type cell is also observed in other small organic molecule electrolysis systems. For example, although high FE up to ~95% at 500 mA cm⁻² is achieved in H-type cell for the E-HAN in alkaline media, the FE decreases to ~80% in the AEM electrolyzer (*Nat. Commun.*, 2023, 14, 3847).

The reasons why the faradaic efficiency for ethylamine production in PEM electrolyzers is lower than that in H-type cell are investigated by considering the unique components in the PEM electrolyzer compared with H-type cell.

On the one hand, the PEM electrolyzer contains special PFSA membrane with high acidic environment during catalysis. To study the influence of local environment on the reaction kinetics, we adjust the electrochemical interfacial interaction between catalyst layer and PFSA membrane by using different membrane electrode assembling method, catalyst-coated membrane (CCM) and catalyst-coated substrate (CCS). In CCM method, catalyst is directly contacted with membrane and exists in highly acidic environment, while in CCS method, catalyst is embedded on PTL and a little farther from the strong acidic environment of PFSA membrane. Based on the results in **Figure R1**, in 0.5 M H₂SO₄ electrolyte containing 8% acetonitrile, the CCM method shows even worse E-HAN performance than the CCS method in terms of the significantly decreased ethylamine FE. Thus, the local environment of PFSA might benefit proton transfer and subsequent electrolysis, resulting in enhanced hydrogen evolution reaction (HER). In comparison, in H-type cell, only minor amount of PFSA is used as binder, which could lead to higher E-HAN FE.

Figure R1. Catalytic performance of the cell prepared by CCM method tested in 0.5 M H₂SO₄ containing 8 wt% acetonitrile solution.

Furthermore, the influence of the carbon-paper based porous transport layer (PTL) in the cathode is studied by performing acetonitrile electroreduction tests in PEM electrolyzers without the cathode catalyst (**Figure R2**). Although carbon is generally supposed to be inactive for HER, in the strongly acidic environment of PFSA, non-negligible current density is observed, indicating the PTL partially contributes to the side reaction in the E-HAN process.

For the mass transport, although the flow cell generally possesses better mass transfer than the H-type cell, in our study, the electrolyte in H-type cell was stirred, which could enhance the mass transfer. Meanwhile, the excess electrolyte, smaller electrode area and low current could ensure local concentration of the reactant. In PEM system, however, the volume of the electrolyte within the electrolyzer is much smaller (~1 mL in our study), both the electrode area the current are much larger, which together lead to quick reactant consumption and decreased local reactant concentration. Thus, despite theoretically better mass transfer in flow cell, the local concentration of the reaction could be even lower than H-type cell, resulting in lower FE of the PEM electrolyzer.

Figure R2. Polarization curve without cathode catalyst in 0.5 M H₂SO₄ containing 8 wt% acetonitrile solution.

In summary, due to the accelerated HER kinetics in the PFSA microenvironment of PEM, the side reaction contributed by gas diffusion layer and the limited mass transfer result in poorer selectivity for ethylamine in PEM compared with the H-type cell.

The comparison between CCS and CCM method have been briefly discussed in the revised manuscript.

3. The investigation into the stability of the Nafion membrane is incomplete. The focus should not only be on membrane dissolution in acetonitrile but also on critical factors like swelling and loss of ion exchange capacity. These aspects are best evaluated under operational conditions of the electrolyzer, not merely by dissolving the membrane in a solution.

Response: We sincerely appreciate the reviewer's insightful suggestion. As suggested, swelling and loss of ion exchange capacity (IEC) of Nafion membrane under operational conditions of the electrolyzer were tested to evaluate its stability.

The swelling of the membrane was detected by measuring its size. In detail, the membrane that were taken out of the electrolyzer after a period of operation time (0-20h) were tested for length in the dry state and length in water and electrolyte for 24 hours each to assess swelling. As shown in Table R1, after only immersing the bare membranes without catalysts into water and the electrolyte, the membrane swelling is observed. However, after loading catalysts onto the membrane, the swelling is reduced due to the adhesion of the catalyst layer. Interestingly, no further membrane swelling is observed under operational condition of 200 mA cm⁻² at various reaction times, indicating the good dimensional and mechanical stability of the membrane during the test (Mater. Res. Bull., 2018, 103, 142–149; Polymer, 2021, 218, 123506).

For the IEC, it could be reflected by the sulfur content in the membrane as the amount of H⁺ released in the sulfonate (-HSO₃) is equal to the amount of S (Solid State Ionics, 2001, 145, 47–51; J. Membrane Sci., 2019, 583, 103–109). Similar to the swelling test, the S element content of the membrane was measured at different reaction times of 200 mA cm⁻² by elemental analysis technique. As displayed in Table R2, the loss of -HSO₃ was not obvious under operational conditions.

In a word, both the mechanical and ion exchange property of the membrane are maintained during the catalysis, demonstrating the stability of the membrane. We have briefly discussed it in the revised manuscript.

Table R1. Swelling ratio of proton exchange membrane at the current density of 200 mA cm⁻².

Sample	Reaction Time	Swelling ratio in H ₂ O	Swelling ratio in 0.5 M H ₂ SO ₄ +8wt% MeCN
Membrane without catalyst	0h	10.2%	18.4%
Membrane with anode catalyst	0h	6.3%	11.6%
Membrane with anode catalyst	5h	5%	12.5%
Membrane with anode catalyst	20h	5.3%	11.8%

Table R2. Number of available exchange sites to proton per gram of Nafion for different reaction time at the current density of 200 mA cm⁻².

Sample	Reaction Time	The proportion of sulfur	Exchange sites per gram (mmol HSO ₃ ⁻ /g)
Membrane with anode catalyst	0h	4.399%	1.37
Membrane with anode catalyst	5h	3.816%	1.19
Membrane with anode catalyst	10h	4.281%	1.34
Membrane with anode catalyst	20h	4.472%	1.4

4. The manuscript lacks a comprehensive economic feasibility comparison of hydrogen oxidation at the anode versus oxygen evolution reaction. The argument that using Pt/C could lower the total costs compared to IrO₂, especially considering the consumption of hydrogen over water, is not substantiated with adequate economic analysis.

Response: We thank the referee for the valuable suggestion. Based on the suggestions of the referee, economic feasibility comparison of hydrogen oxidation at the anode versus oxygen evolution reaction has been investigated. Techno-economic analysis reveals that at a typical commercial current densities (~200 mA cm⁻²), the OER||HAN yields a negative upcycling income of approximately \$-85.25 per ton of acetonitrile (**Figure R3a**). In contrast, when the anode is engaged in the hydrogen oxidation reaction (HOR), factoring in hydrogen consumption, the upcycling income jumps to around \$261.66 per ton. This indicates that, under industrial current conditions, the economic benefits of acetonitrile reduction in conjunction with HOR significantly outweigh those of OER (**Figure R3b**). We have incorporated the suggestions made by the reviewer into the revised manuscript and have added corresponding discussions. Additionally, **Figure R3** have been included in the Supporting Information as **Supplementary Fig. 16**.

Figure R3. Techno-economic analysis (TEA) of the electrocatalytic hydrogenation of acetonitrile to ethylamine by anodic oxygen evolution or hydrogen oxidation reaction at 200 mA cm⁻². (a) Oxygen evolution reaction at the anode. (b) Hydrogen oxidation reaction at the anode.

5. The interpretation of SR-FTIR data, particularly the vibration band intensities of CCN and CH₃ of ethylamine, is not accurate and insightful. The strong signals corresponding to ethylamine on Pd, compared to Cu, Pt, and Au, are expected due to Pd's efficacy in acidic acetonitrile electroreduction. This observation does not provide mechanistic insights into why Pd is the preferred catalyst for acetonitrile reduction in acid.

Response: We deeply thank the referee for the valuable comment and guidance. According to the reviewer's opinion, we have re-analyzed the significance of the changes in vibration band intensities. The vibration mode of CCN is attributed to the $\delta(\text{C-C}\equiv\text{N})$, which is one of the characteristic vibration peaks of acetonitrile (Electrochim. Acta. 1996, 41, 641–651). Since the peak intensity in SR-FTIR is also related to the molecular coverage (ACS Catal. 2021, 11, 2473–2482), the increased acetonitrile signal intensity on Pd surface unveils increased acetonitrile surface coverage when potential sweeps negatively. In comparison, the $\delta(\text{C-C}\equiv\text{N})$ band intensity changes on Cu, Pt, and Au are much less significant than that on Pd, indicating that lower acetonitrile molecule coverage on these metal surfaces. Generally, higher molecular coverage could benefit the reaction kinetics (Mol. Catal., 2021, 504, 111482; Nat. Catal., 2020, 3, 775–786), which may contribute to the enhanced activity of Pd in acidic E-HAN.

Based on DFT calculations and OCP tests, Pd surface possesses stronger affinity towards acetonitrile than Cu, Ag and Au, which could explain the higher acetonitrile coverage on Pd than on these metals. Although Pt shows theoretically close adsorption strength of acetonitrile with Pd, the more beneficial HER kinetics on Pt leads to diminished competitive adsorption of acetonitrile. Thus, combining the SR-FTIR results, OCP tests and DFT calculations, the more preferable acetonitrile adsorption could be responsible for the highest acetonitrile reduction activity on Pd surface.

In summary, while the manuscript addresses a topic of potential interest, these significant gaps in rationale, experimental design, and data interpretation prevent me from recommending its publication in its current state.

Response: We sincerely thank the referee for carefully reviewing our manuscript and recognizing the topic of our study is interesting. We also deeply appreciate the valuable comments and guidance which certainly help to improve our manuscript. According to the reviewer's suggestions, we have reexplained the microenvironmental differences between the H-type cell and the membrane electrode, evaluated the stability of the membrane under operational conditions, and reinterpreted the SR-FTIR data. We hope that with these significant improvements and clarifications, the manuscript can now be recommended for publication.

To Reviewer #2:

The authors have answered all questions raised by the reviewers. The reviewers appreciate the authors' dedication to this revision effort. Most additional data and responses were adequate. There are still potential challenges to overcome in the proposed E-HAN process, notably the very low conversion of starting materials (only 1.35% in the single-pass experiment). The reviewers believe that this value is too low to imagine real-world applications. Even so, this paper demonstrates the high potential of the PEM system as an E-HAN process, which gives important insights into electrifying the current energy-consuming thermal processes.

After the revision, this manuscript is in good shape and meets the high criteria and broad interest of Nature Communications. Thus, the reviewer recommends the acceptance of the manuscript for publication.

Response: We deeply thank the reviewer's comments. We agree with the reviewer's concerns about low conversion of starting materials in the single-pass experiment. We think conversion could be greatly improved by recycling the electrolyte solution. This manuscript focuses on the first attempt to select the catalysts and study the mechanism of acidic E-HAN process. The conversion limitation raised by the reviewer is very meaningful, and we will conduct a systematic study in the future.

To Reviewer #3:

The authors have modified the manuscript according to the reviewer's advice, but I still think this study is very reluctant due to the low FE (43.8%) in the acidic condition of PEM and use the noble metal catalysts compared to the Cu catalyst in alkaline condition, and from the revised supporting information in Figure R10, we can know that the economy is not very good although the authors think the acidic E-HAN has its own advantages on product separation.

Response: We deeply thank the reviewer's comments and we are pleased to clarify reviewer's concerns. This is the first attempt at E-HAN under acidic conditions, and various in-situ characterization methods have been used to understand the reaction mechanism. This article broadens the research interest of acetonitrile reduction reaction and should belong to fundamental research. With further improvements in the catalyst

and electrolytic cell structure, the overall performance of the device may be further enhanced, enabling it achievable to E-HAN performance in alkaline conditions.

REVIEWER COMMENTS

Reviewer #1 (Remarks to the Author):

After a thorough examination of the rebuttal letter and the revised manuscript, I acknowledge that the authors have addressed most of the initial concerns, enhancing the overall quality of the paper.

Nevertheless, there remain a few minor issues that require attention:

1. In the economic analysis comparing acetonitrile hydrogenation paired with oxygen evolution reaction and hydrogen oxidation reaction, the cost of hydrogen is a significant factor that is not currently listed. A credible source for this cost must be provided. Furthermore, it should be clarified that pairing hydrogen oxidation, particularly when the hydrogen is derived from fossil fuels, with electrochemical manufacturing processes would result in a net increase in carbon emissions, which contradicts the aim of achieving a carbon-negative footprint.
2. The comparison of peak intensities in SR-FTIR data between samples is not justifiable. Specifically, the increased signal intensity of acetonitrile on Pd compared to Cu does not directly indicate a higher surface coverage of acetonitrile on Pd. The authors should revise their interpretation or provide additional evidence to substantiate their claim.

Response to Reviewers

We sincerely appreciate you for spending the valuable time evaluating our manuscript and giving us constructive suggestions to improve the quality of the manuscript. The point-by-point responses to your comments are attached below and all the corresponding revisions newly made are highlighted using yellow in the revised manuscript.

To Reviewer #1:

After a thorough examination of the rebuttal letter and the revised manuscript, I acknowledge that the authors have addressed most of the initial concerns, enhancing the overall quality of the paper. Nevertheless, there remain a few minor issues that require attention:

Response: We deeply thank you for the positive feedback on our work and the valuable comments, which help us improve the quality of the manuscript. According to the your suggestions, we have carefully revised our manuscript.

1. In the economic analysis comparing acetonitrile hydrogenation paired with oxygen evolution reaction and hydrogen oxidation reaction, the cost of hydrogen is a significant factor that is not currently listed. A credible source for this cost must be provided. Furthermore, it should be clarified that pairing hydrogen oxidation, particularly when the hydrogen is derived from fossil fuels, with electrochemical manufacturing processes would result in a net increase in carbon emissions, which contradicts the aim of achieving a carbon-negative footprint.

Response: We sincerely appreciate your insightful suggestion and comment. As suggested, the cost of hydrogen have been listed in **Supplementary Table 1**. And we admit that pairing hydrogen oxidation, especially when the hydrogen is sourced from fossil fuels, with electrochemical manufacturing processes would result in a net increase in carbon emissions. To achieve a carbon-negative footprint, it is crucial to use hydrogen that is produced from renewable sources, such as electrolysis using electricity from wind or solar power. The renewable hydrogen, often referred to as “green hydrogen” does not emit carbon during its production and can therefore be used in electrochemical manufacturing without contradiction to the aim of reducing carbon emissions. Pairing hydrogen oxidation with electrochemical manufacturing processes is increasingly attractive as the cost of green hydrogen is being reduced.

Anyway, this paper focuses on screening catalysts for the reduction of acetonitrile in acidic conditions. Since the issue of carbon emission is not the original intention of our study, we have not discussed it extensively.

2. The comparison of peak intensities in SR-FTIR data between samples is not justifiable. Specifically, the increased signal intensity of acetonitrile on Pd compared to Cu does not directly indicate a higher surface coverage of acetonitrile on Pd. The authors should revise their interpretation or provide additional evidence to substantiate their claim.

Response: Thanks for the valuable comment and guidance. We apologize for the inappropriate understanding of the peak intensity change. Recognizing the constraints of infrared intensity analysis, we have amended our statement to reflect that we observe that acetonitrile molecules accumulate to varying extents on different metal surfaces, and this variation is probably associated with the differing strengths of molecular adsorption on these surfaces. To more quantitatively evaluate the adsorption behavior over different catalyst, the open-circuit potential (OCP), reflecting adsorbates in the inner Helmholtz layer, is performed. As displayed in Fig. R1 (Figure S23), moderate adsorption strength of MeCN molecules on Pd surfaces is inferred, which may also be the reason for its high activity in acidic E-HAN. We have revised their interpretation in the revised manuscript.

Figure R1. OCP of Pt/C (a), Pd/C (b), Cu/C (c), Au/C (d) and Ag/C (e) in 0.5 M H₂SO₄ solution before and after 8% MeCN was added.

REVIEWERS' COMMENTS

Reviewer #1 (Remarks to the Author):

Upon reviewing the revised manuscript, I am satisfied that all previous comments have been thoroughly addressed. However, I suggest removing the discussion on open-circuit potential (OCP) as a measure of absorbing energy, as it may not be adequately justified.

Given the improvements and pending the suggested adjustment, I recommend the manuscript for publication without further review.

To Reviewer #1:

Upon reviewing the revised manuscript, I am satisfied that all previous comments have been thoroughly addressed. However, I suggest removing the discussion on open-circuit potential (OCP) as a measure of absorbing energy, as it may not be adequately justified.

Given the improvements and pending the suggested adjustment, I recommend the manuscript for publication without further review.

Response: We deeply appreciate the reviewer's dedication in assessing our manuscript and are grateful for the positive feedback on our research. Following the reviewer's suggestion, the description of the open-circuit potential (OCP) has been omitted from the revised manuscript.